# Partial freezing of rat livers extends preservation time by 5-fold

Shannon N. Tessier [1,2,9], Reinier J. de Vries[1,2,3,9], Casie A. Pendexter [1,2,8], Stephanie E. J. Cronin[1,2], Sinan Ozer[1,2], Ehab O. A. Hafiz [4], Siavash Raigani [2,5], Joao Paulo Oliveira-Costa[1,6], Benjamin T. Wilks[1,2], Manuela Lopera Higuita[1,2], Thomas M. van Gulik[3], Osman Berk Usta[1,2], Shannon L. Stott[1,6], Heidi Yeh[5], Martin L. Yarmush[1,2,7], Korkut Uygun [1,2✉] & Mehmet Toner [1,2✉]

The limited preservation duration of organs has contributed to the shortage of organs for transplantation. Recently, a tripling of the storage duration was achieved with supercooling, which relies on temperatures between −4 and −6 °C. However, to achieve deeper metabolic stasis, lower temperatures are required. Inspired by freeze-tolerant animals, we entered high-subzero temperatures (−10 to −15 °C) using ice nucleators to control ice and cryoprotective agents (CPAs) to maintain an unfrozen liquid fraction. We present this approach, termed partial freezing, by testing gradual (un)loading and different CPAs, holding temperatures, and storage durations. Results indicate that propylene glycol outperforms glycerol and injury is largely influenced by storage temperatures. Subsequently, we demonstrate that machine perfusion enhancements improve the recovery of livers after freezing. Ultimately, livers that were partially frozen for 5-fold longer showed favorable outcomes as compared to viable controls, although frozen livers had lower cumulative bile and higher liver enzymes.

[1] Center for Engineering in Medicine and Surgery, Harvard Medical School and Massachusetts General Hospital, Boston, MA, USA. [2] Shriners Hospitals for Children Boston, Boston, MA, USA. [3] Department of Surgery, Amsterdam University Medical Centers – location AMC, University of Amsterdam, Amsterdam, the Netherlands. [4] Department of Electron Microscopy Research, Theodor Bilharz Research Institute, Giza, Egypt. [5] Department of Surgery, Division of Transplantation, Massachusetts General Hospital, Boston, MA, USA. [6] Department of Medicine and Cancer Center, Massachusetts General Hospital, Harvard Medical School, Charlestown, MA, USA. [7] Department of Biomedical Engineering, Rutgers University, Piscataway, NJ, USA. [8] Present address: Sylvatica Biotech Inc., North Charleston, SC, USA. [9] These authors contributed equally: Shannon N. Tessier, Reinier J. de Vries. ✉email: KUYGUN@mgh.harvard.edu; mehmet_toner@hms.harvard.edu

The need for transplantation is growing steadily while the supply of donor organs is nowhere near demand[1,2]. Extending organ preservation has increasingly been identified as a national research priority[3–5] that would significantly impact organ allocation, handling, and transplantation in several important ways. Firstly, for livers that can currently reach their recipient using clinical hypothermic preservation, extending preservation duration would convert from emergency to planned surgeries, reduce the cost of transplantation, and enable improved matching according to HLA compatibility. Secondly, immune tolerance induction protocols[6,7] are poised to eliminate rejection, thereby increasing the quality of life of recipients, extending in vivo graft life, and reducing the need for re-transplantation. However, these protocols may require more time to prepare the recipient than is currently possible with clinical preservation and hence extended preservation will be required to fully realize this breakthrough achievement. Thirdly, some organs that are procured for transplantation are discarded due to circumstantial factors that include donor/recipient location. These factors could be eliminated with extended preservation duration. Finally, when complimentary fields such as tissue engineering and regenerative medicine achieve future breakthroughs in artificial organ engineering, improved organ preservation methods will be required to enable off-the-shelf access to these life-saving organs[1,8]. However, it should be noted that to fully transform organ allocation, utilization, and transplantation practices, an essential compliment to extensions of preservation times will be development of reliable organ assessment tools.

Preservation methods diverge in two strategies to slow down deterioration of donor organs outside the human body: metabolic support and metabolic depression. Each approach has advantages and disadvantages (e.g., length of preservation, thermodynamic stability, ease of operation, accessibility, etc.) that will ultimately necessitate a head-to-head comparison of all available technologies prior to clinical translation. Metabolic support through machine perfusion can resuscitate injured liver grafts, assess degree of injury, and moderately extend the preservation time[1,9–12]. Despite these advantages of resuscitation and assessment, technology to maintain ex vivo homeostasis becomes more complex with increasing perfusion durations and requires continuous monitoring and adaptations to the system[8,12]. Metabolic depression takes advantage of the fact that the metabolic rate – and consequently tissue deterioration – slows down at decreasing temperatures[13]. The clinical standard for organ preservation is hypothermic preservation at +4 °C. However, this limits storage for vascular and metabolically active tissues such as the liver to the order of hours, with 9 h being the typical clinical preservation duration for liver transplantation. Decreasing the storage temperature below freezing holds great promise to depress metabolism and extend preservation durations beyond clinical standards; however, major challenges in the preservation of complex tissues at subzero temperatures have proven difficult to overcome. Importantly, an optimal preservation protocol may not need to choose one strategy versus the other since the advantages of machine perfusion in resuscitation and assessment can be combined with periods of deep metabolic stasis. Our past efforts have already demonstrated this synergy, whereby we used machine perfusion to recover rodent[14,15] and human[13,16] livers after a period of subzero metabolic rate depression to extend preservation duration. We also aim to demonstrate this important synergy between machine perfusion and subzero preservation in the present work.

Most subzero preservation efforts have focused on low cryogenic temperature ranges (<−80 °C)[1]. At these temperatures, freezing or vitrification approaches can suffer from either lethal intracellular ice formation, mechanical and thermal stresses, cryoprotectant toxicity and/or limited scalability from cell to human organ-sized systems[17,18]. Instead, there is a potentially advantageous – yet relatively unexplored – "high subzero" temperature range from −4 °C to −20C which may enable deeper metabolic stasis than clinical hypothermic storage at +4 °C, while avoiding many of the above-mentioned challenges of deep cryogenic storage[19,20]. We previously showed that ice formation in rodent and human livers can be completely avoided at high subzero temperatures (−4 °C to −6 °C) with supercooling preservation, resulting in a tripling of the preservation duration of mammalian livers (3 days in rodents and 27 h in humans)[13–15]. While retaining the preservation media in the liquid state circumvented challenges in phase changes of water that have eluded cryopreservation scientists for decades[21–23], the ice-free supercooled state is thermodynamically unstable and always at risk for spontaneous ice formation (which is far more injurious than equilibrium freezing). Since the risk of ice formation increases as the storage temperature decreases, supercooling is inherently limited by the depth of metabolic stasis that can be achieved. Yet, extension of preservation duration of human livers beyond 27 h would be required to enable global matching programs. Taken together, alternative strategies will be required to reach lower storage temperatures and even longer preservation durations.

Freeze tolerance is an effective strategy utilized by multiple organisms in the nature[24]. Wood frogs (Rana sylvatica) can survive in a frozen state at −6 °C to −16 °C[25] for weeks[26,27]. The wood frog capitalizes on both ice nucleating agents (INA) and endogenous CPAs to orchestrate freezing and prevent injurious intracellular ice formation. The INAs promote ice formation within the vasculature as close as possible to melting point and studies in freeze-tolerant species showed that controlled freezing of extracellular water by INAs is critical for freezing survival[28]. As extracellular water gradually freezes, it is accompanied by an increase in the osmolality of the non-frozen extracellular fluid. This results in cellular dehydration as water is pulled from the intracellular environment[26,27]. Another important strategy that confer freeze-tolerance is the synthesis of high amounts of carbohydrates, such as glucose in wood frogs[29–31]. Glucose in the blood and tissues provides colligative resistance to detrimental decreases in cell volume and together with INAs restricts the formation of intracellular ice.

Adopting some of these learnings, we present a protocol for freezing of whole rat livers that enters high subzero temperatures ranging from −10 °C to −15 °C for durations of up to 5 days in the presence of ice. With a focus on conferring a non-injurious frozen state, we challenge a central paradigm in cryopreservation that ice should be completely avoided. Our approach leverages CPAs inspired from freeze-tolerant wood frogs such as INAs and the glucose analog, 3-O-methyl-D-glucose (3-OMG). Additionally, we compare the effectiveness of classical permeating CPAs, such as glycerol (GLY), ethylene glycol (EG), and propylene glycol (PG), and evaluate outcomes based on storage temperatures (−10 °C and −15 °C) and duration of storage (1 and 5 days). We further bolster our approach by leveraging machine perfusion technology to address challenges in loading of high osmolality CPAs and key engineering principles that are critical to the recovery/assessment of livers after freezing. The central strategy of this protocol is to promote a thermodynamically stable frozen state, while maintaining a sufficient unfrozen fraction to limit ice damage and excessive dehydration; therefore, we coined this approach "partial freezing".

## Results

**Partial Freezing protocol optimization.** Our initial efforts incorporated nature-inspired INAs (Snomax 1 g/L), 3-OMG

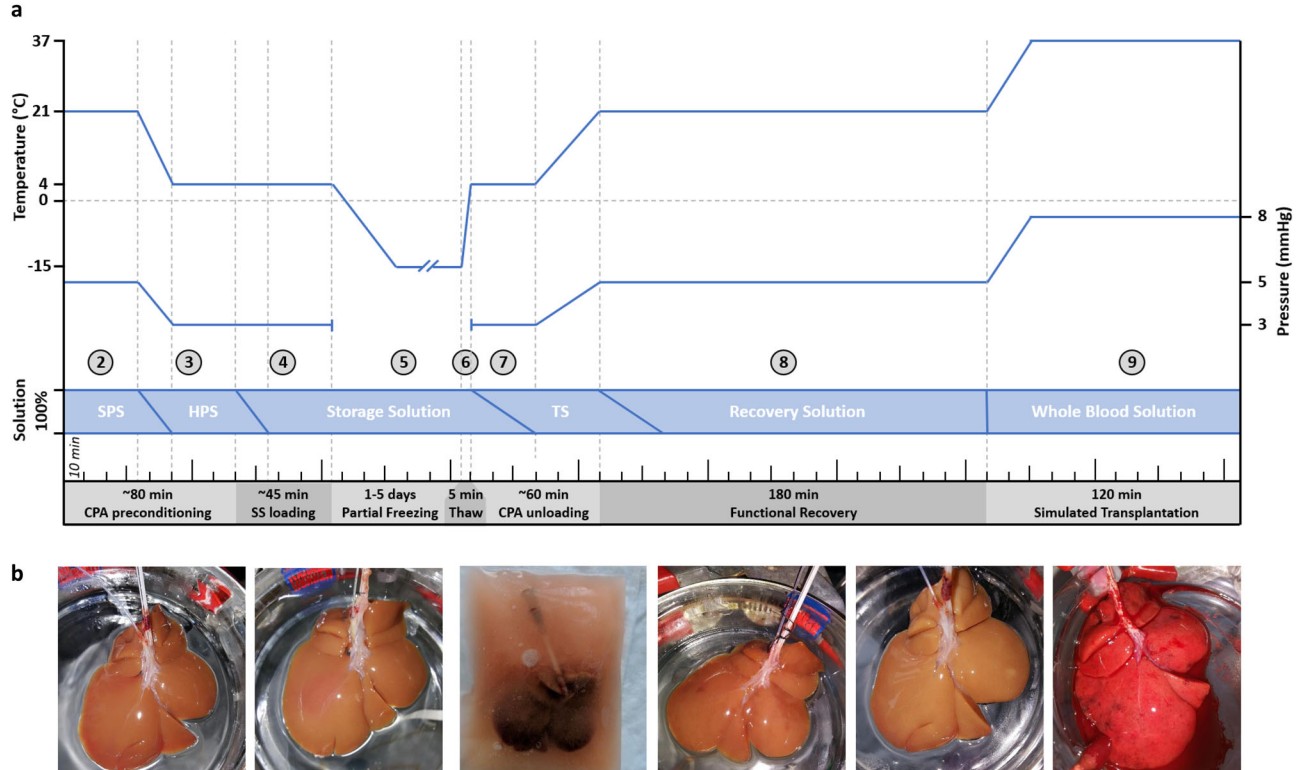

**Fig. 1 The partial freezing protocol. a** Schematic overview of the partial freezing protocol showing the target perfusion temperature (top blue line) and pressures (bottom blue line) during the subsequent steps in the protocol. Livers were stored at either −10 or −15 °C. The numbers in circles match the explanation of the protocol in the introduction and Methods section. The perfusion solutions and the rate of change between the solutions are shown in blue boxes. SPS = subnomothermic preconditioning solution, HPS = hypothermic preloading solution, TS = thawing solution. See Supplementary Table S1 for the exact composition of the solutions. **b** Photos of the livers during the consecutive steps of the protocol. From left to right; SNMP preloading, HMP preloading, partial freezing, CPA unloading, functional recovery during SNMP, and start of simulated transplantation.

(200 mM), and PEG (5%) in combination with the classical cryoprotectant glycerol (6% vol/vol) into the storage solution and stored livers at −6 °C in the frozen state. The storage solution was loaded and unloaded in one step with a syringe by hand and the livers were thawed in a warm (37 °C) water bath with perfusate. However, this method resulted in excessive edema (>60%) and high vascular resistances (mean 0.11 ± 0.05 mmHg min/ml g; mean ± SD throughout the text, unless otherwise specified) during SNMP recovery after freezing (Supplementary Fig. S1a, b). Gross morphological assessment indicated inadequate perfusion of the periphery, as shown on Supplementary Fig. S1c, d, which was likely caused by the incomplete delivery of CPAs and dramatic shifts in osmolality during (un)loading, that were further aggravated by frozen storage.

To improve these results, we modified our approach by (1) pressure- and temperature-controlled loading/unloading during machine perfusion instead of a hand syringe flush; (2) multi-step loading of CPAs with gradual changes between the different solutions; (3) higher concentration of permeating CPAs (12% instead of 6%), (4) addition of osmotic counterbalancing non-permeating CPAs (trehalose and raffinose); and (5) the use of glutathione during SNMP recovery. Taken together, these improvements resulted in significantly less edema, lower vascular resistance and improved flow rates during machine perfusion after freezing (see results of Supplementary Fig. S1). Ultimately, this optimized protocol (presented in Fig. 1 and Supplementary Table S1) is described in detail in the "Methods". The final protocol entails 9 consecutive steps: (1) procurement of the liver (2) preconditioning during subnormothermic machine perfusion (SNMP), (3) preloading of CPAs during hypothermic machine

perfusion (HMP), (4) loading of the final storage solution during HMP, (5) freezing, (6) thawing, (7) unloading of CPAs during HMP, (8) functional recovery during SNMP, and (9) viability assessment during ex vivo simulated transplantation.

**Effect of permeating cryoprotective agents and warm ischemic injury on liver viability after partial freezing.** Using the protocol presented in Fig. 1, we compared the effects of glycerol (GLY), ethylene glycol (EG) and propylene glycol (PG) on liver viability after partial freezing at −10 °C for 1 day to a control group of conventional hypothermic preservation for 1 day at +4 °C. Previous studies have repeatedly shown this 1 day duration of hypothermic preservation results in 100% transplant survival in the same animal model[14,32,33]. While SNMP after freezing provides the opportunity to assess liver function/injury between the experimental groups[14,34,35], we also simulated transplantation with ex vivo normothermic reperfusion in the presence of whole blood. This is an established model to simulate transplantation that has been used in rat, swine and human livers[36–40]. While we will focus on the viability assessment during simulated transplantation (Figs. 2–5, Supplementary Figs. S2, S3), parameters of liver function and injury during SNMP are provided in the Supplementary materials (Supplementary Figs. S4–S6).

Although edema was substantially improved with gradual CPA loading/unloading, GLY livers still gained considerable weight during SNMP recovery (Supplementary Fig. S4a; 35.00 ± 20.49% at 3 h) and the final weight gain during simulated transplantation was nonetheless 19.00 ± 16.63% at 2 h (Fig. 2a). Conversely, the weight of the EG and PG livers as well as the controls was stable during simulated transplantation (2.50 ± 5.26%; and 4.00 ± 9.20%

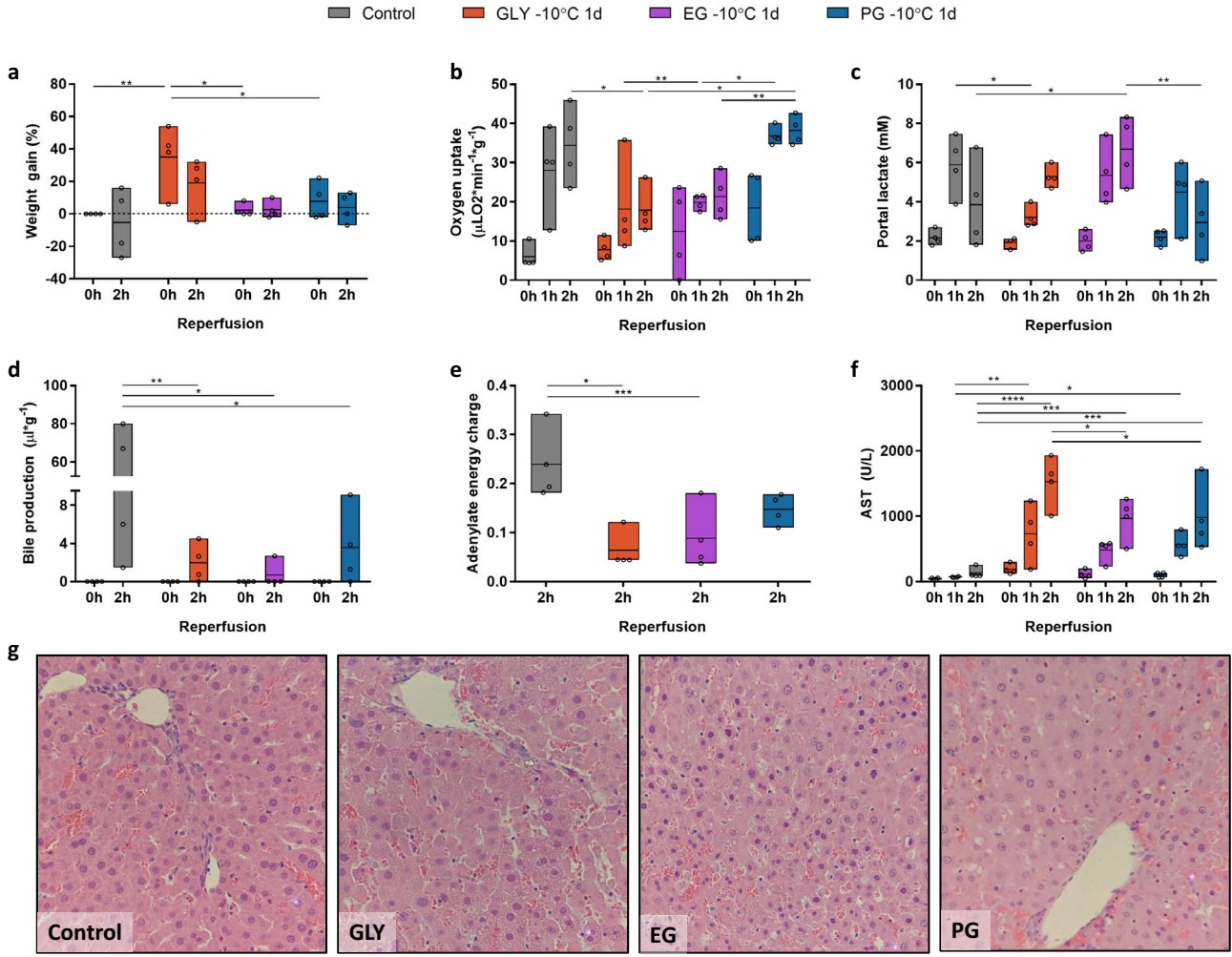

**Fig. 2 Effect of permeating cryoprotective agents on liver function, injury, and microscopic tissue structure after partial freezing at −10 °C for 24 h.**
**a** Weight gain as percentage of the procurement weight. **b** Oxygen uptake. **c** Lactate concentration in the portal vein. **d** Bile production. **e** Tissue adenylate energy charge. **f** Aspartate aminotransferase concentration (AST) in the intrahepatic vena cava (IVC). **g** Light microscopy images of parenchymal liver wedges at the end of simulated transplantation (×40). Controls (gray) = 1 day hypothermic preservation, *GLY* glycerol (red), *EG* ethylene glycol (purple), *PG* propylene glycol (dark blue). Stars denote statistical significance (two-way ANOVA, followed by Tukey's post-hoc test): *$0.01 < p < 0.05$; **$0.001 < p < 0.01$; ***$0.0001 < p < 0.001$; ****$p < 0.0001$. Boxes: floating bars (min to max), with a line at the mean. Source data are provided as a Source Data file.

at 2 h, respectively). Of the three experimental groups, the vascular resistances of the PG livers were most comparable to the controls during simulated transplantation ($0.019 ± 0.011$ *vs* $0.016 ± 0.006$ mmHg min mL$^{-1}$ g$^{-1}$ at 2 h, respectively, Fig S2a).

It has been previously shown that oxygen uptake during SNMP recovery after supercooling (subzero non-frozen preservation) was significantly correlated to transplant survival in rat livers[14]. The oxygen uptake during simulated transplantation (Fig. 2b) of the PG livers ($38.22 ± 3.66$ μLO$_2$ min$^{-1}$ g$^{-1}$ at 2 h) was the same as control livers ($34.45 ± 9.91$ at 2 h) while the oxygen uptake of the GLY livers ($17.85 ± 5.86$ at 2 h) and EG livers ($21.41 ± 5.80$ at 2 h) was significantly reduced compared to the controls ($p = 0.0186$ and $p = 0.0879$, respectively) and PG livers ($p = 0.0028$ and $p = 0.0169$, respectively). A similar trend was observed in portal lactate, which is an important parameter of liver function[41]. While the lactate concentration increased for all livers during the first hour of simulated transplantation (Fig. 2c), control and PG livers subsequently cleared lactate and this was not observed for GLY and EG livers. We also assessed the effect of warm ischemia time (WIT) on perfusion outcomes after freezing to ascertain if our protocol would be compatible with livers that had already undergone mild to

moderate injury. Using PG as the main permeating CPA, Supplementary Figure S7 compares livers that were exposed to 30 min warm ischemia prior to freezing at −10 °C for 1 day versus livers frozen under the same conditions without exposure to warm ischemic injury. At the end of simulated transplantation (2 h), the results show that there were no statistically significant changes in all parameters measured, except for lactate (WIT was $6.37 ± 1.09$ mM versus PG $2.95 ± 1.73$ mM; $p = 0.0018$).

Bile production is another important metric of liver function. Most GLY and EG livers did not produce bile during simulated transplantation (Fig. 2d). Conversely all but one PG livers did produce bile; however, the total cumulative volume was significantly lower compared to controls ($3.56 ± 4.02$ *vs* $38.61 ± 40.66$ μl g$^{-1}$ at 2 h; $p = 0.0111$). Tissue adenylate energy content is considered a representative metric for liver viability[35,42–45]. While all livers that were stored in the frozen state at −10 °C for 1 day showed a decreasing trend of tissue adenylate energy charge (Fig. 2e), statistically significant decreases were only observed for GLY ($0.064 ± 0.039$ at 2 h) and EG ($0.088 ± 0.065$ at 2 h), as compared to control livers ($0.24 ± 0.073$ at 2 h; $p = 0.0033$ and $p = 0.0097$, respectively).

In addition to parameters of liver function, we assessed general and specific parameters of (hepato)cellular injury during simulated transplantation. Increased potassium concentrations are a general parameter of cellular injury as intracellular potassium is released after cell death. However, the release of the liver specific enzymes aspartate aminotransferase (AST) and alanine aminotransferase (ALT) are more specific to hepatocellular injury. All three parameters showed indications of increased injury after partial freezing in all experimental groups compared to the controls (see Fig. 2f, Supplementary Fig. S2b, c); however, the AST levels of the GLY livers ($1528 \pm 388$ U/L at 2 h) were significantly higher than the EG livers ($967 \pm 330$ at 2 h) and PG livers ($979 \pm 520$ at 2 h; $p = 0.0207$ and $p = 0.0244$, respectively). Finally, microscopic tissue structure was assessed using hematoxylin and eosin-stained liver slices by a blinded pathologist, with representative brightfield images presented in Fig. 2g. Livers that were stored with GLY showed the most unfavorable results with severe loss of LSECs, which agrees with perfusion metrics reported above. In contrast, favorable to modest results were observed for livers stored in the presence of EG and PG.

**Effects of storage temperature and duration on liver viability after partial freezing.** Livers stored with PG and EG clearly outperformed GLY livers. While EG livers showed comparable if not slightly favorable results upon histological analysis, PG showed higher viability in most perfusion parameters described above. For these reasons, we used PG as the main permeating CPA to test the effect of freezing temperature ($-10\,°C$ vs $-15\,°C$) and storage duration (1 vs 5 days) on viability after partial freezing.

While there was no statistically significant change in weight gain during simulated transplantation of 1-day frozen livers at $-10$ and $-15\,°C$ (Fig. 3a), the increase in storage duration from 1 to 5 days resulted in significantly higher weight gain during SNMP recovery after freezing (Supplementary Fig. S5a; $40.00 \pm 4.32\%$ at 3 h) that did not normalize during simulated transplantation (Fig. 3a; $30.25 \pm 5.56\%$ at 2 h). Vascular resistance (Supplementary Fig. S3a) was not significantly different after freezing at $-10\,°C$ and $-15\,°C$ ($0.019 \pm 0.011$ vs $0.021 \pm 0.010$ mmHg min mL$^{-1}$ g$^{-1}$ at 2 h during simulated transplantation, respectively) as well as between livers frozen for 1 and 5 days ($0.011 \pm 0.004$ vs $0.021 \pm 0.021$ mmHg min mL$^{-1}$ g$^{-1}$ at 2 h, respectively).

Although vascular resistance across all groups were comparable, parameters of liver function were negatively affected by the lower freezing temperature, yet unchanged by the increase in storage duration. The oxygen uptake during simulated transplantation (Fig. 3b) was significantly reduced after freezing at $-15\,°C$ for 1 day compared to $-10\,°C$ for 1 day ($23.47 \pm 4.68$ vs $36.86 \pm 2.34$ at 1 h; $24.85 \pm 2.38$ vs $38.22 \pm 3.66$ $\mu$LO$_2$ min$^{-1}$ g$^{-1}$ at 2 h; $p = 0.0443$ and $p = 0.0447$, respectively), although not markedly different when comparing freezing at $-15\,°C$ for 1 vs 5 days ($24.85 \pm 2.38$ vs $25.35 \pm 9.78$ $\mu$LO$_2$ min$^{-1}$ g$^{-1}$ at 2 h). Similarly, livers frozen at $-15\,°C$ for 1 day did not clear lactate during simulated transplantation while livers frozen at $-10\,°C$ for 1 day did (Fig. 3c), although differences between the lactate levels at the discrete time points did not reach statistical significance. When comparing livers frozen to $-15\,°C$ for 1- and 5 days, no lactate clearance was observed and lactate levels at the end of simulated transplantation were comparable ($5.21 \pm 2.67$ vs $5.73 \pm 1.29$ mM at 2 h, respectively).

Even after freezing for 5 days at $-15\,°C$, livers produced bile during simulated transplantation (Fig. 3d) and this was not significantly different in livers that were frozen for 1 day at $-15\,°C$ ($3.32 \pm 2.43$ vs $6.11 \pm 4.69$ $\mu$l g$^{-1}$) or 1 day at $-10\,°C$ ($3.32 \pm 2.43$ vs $3.56 \pm 4.02$ $\mu$l g$^{-1}$); however, we note that the

production after freezing for 5 days was significantly lower than controls ($38.61 \pm 40.66$ $\mu$l g$^{-1}$, $p = 0.0111$). In terms of tissue adenylate energy charge (Fig. 3e), all experimental groups showed a decreasing trend in adenylate energy charge as compared to controls ($0.24 \pm 0.073$), although only livers stored at $-15\,°C$ for 1 day ($0.061 \pm 0.019$; $p = 0.0004$) and 5 days ($0.062 \pm 0.029$, $p = 0.0004$) reached statistical significance.

With respect to markers of hepatocellular injury, the AST (Fig. 3f) levels were significantly higher after freezing at $-15\,°C$ for 1 day compared to after freezing at $-10\,°C$ for 1 day ($1283 \pm 224$ vs $567 \pm 172$ U/L at 1 h and $1995 \pm 530$ vs $979 \pm 521$ U/L at 2 h; $p = 0.0035$ and $p < 0.0001$, respectively) and this same trend was also evident in the ALT levels (Supplementary Fig. S3c). The maximum potassium (Supplementary Fig. S3b) and ALT values (Supplementary Fig. S3c) during simulated transplantation after 1 and 5 days freezing at $-15\,°C$ were the same (potassium = $5.05 \pm 0.48$ vs $4.95 \pm 0.48$ mM at 2 h; ALT = $2335 \pm 597$ vs $2316 \pm 242$ U/L at 2 h, respectively). Finally, analysis of microscopic tissue structure (Fig. 3g) showed livers stored at both $-10\,°C$ and $-15\,°C$ for 1 day showed some disruption of the lobular architecture with intact portal triads and hepatocyte swelling, although livers stored at $-15\,°C$ showed more loss of LSECs. For livers stored at $-15\,°C$ for 5 days, the central veins also showed early subendothelial congestion and endothelial cell disruption with moderate to marked loss of LSECs.

**Effects of a clinical grade oxygenator on liver viability after partial freezing.** While normothermic temperatures and the presence of whole blood are essential to fully realize tissue injury, recirculating whole blood may interact with the surfaces of the perfusion system, thereby triggering coagulation and inflammatory events that are not reflective of in vivo events. To address this, we substituted a clinical grade oxygenator (Affinity Pixie Oxygenator) that contains a hydrophilic polymer coating to reduce platelet adhesion and activation. In Figs. 4–5 and S6, S8, 9, we present perfusion outcomes during a 2-h simulated transplantation that compare control data (1-day hypothermic preservation) using the Radnoti versus Affinity Pixie Oxygenator (denoted as "Clinical Oxy. Control"), as compared to livers that were partially frozen in the presence of propylene glycol (PG) at $-15\,°C$ for 1 and 5 days with the Affinity Pixie Oxygenator (denoted as "Clinical Oxy. PG").

Liver weight gain during simulated transplantation (Fig. 4a) was not significantly different for 1-day livers ($12.00 \pm 8.66\%$ at 2 h) but significantly elevated for 5 days livers ($26.00 \pm 15.13\%$ at 2 h; $p = 0.020$), as compared to Clinical Oxy. Controls ($-4.67 \pm 5.13\%$ at 2 h). Vascular resistance (Fig. 4b) was not significantly different after freezing at $-15\,°C$ for either 1 day ($0.004 \pm 0.002$ mmHg min mL$^{-1}$ g$^{-1}$ at 2 h) or 5 days ($0.007 \pm 0.010$ mmHg min mL$^{-1}$ g$^{-1}$ at 2 h), as compared to Clinical Oxy. Controls ($0.010 \pm 0.001$ mmHg min mL$^{-1}$ g$^{-1}$ at 2 h) for all time points during simulated transplantation. By the end of simulated transplantation, parameters of liver function, including oxygen uptake (Fig. 4c) and portal lactate (Fig. 4d), were similar with no statistically significantly differences observed across all experimental groups and controls. However, bile production (Fig. 4e) was significantly reduced when comparing 1 day ($0.67 \pm 0.92$ $\mu$l g$^{-1}$ at 2 h) and 5 days livers ($2.92 \pm 3.74$ $\mu$l g$^{-1}$ at 2 h) partially frozen at $-15\,°C$, as compared to the Clinical Oxy. Controls ($62.02 \pm 51.72$ at 2 h). Despite a fivefold longer storage, livers that were partially frozen for 5 days at $-15\,°C$ ($0.131 \pm 0.031$) did not show a statistically significant difference in tissue adenylate energy charge as compared to Clinical Oxy. Controls ($0.242 \pm 0.084$; Fig. 4f).

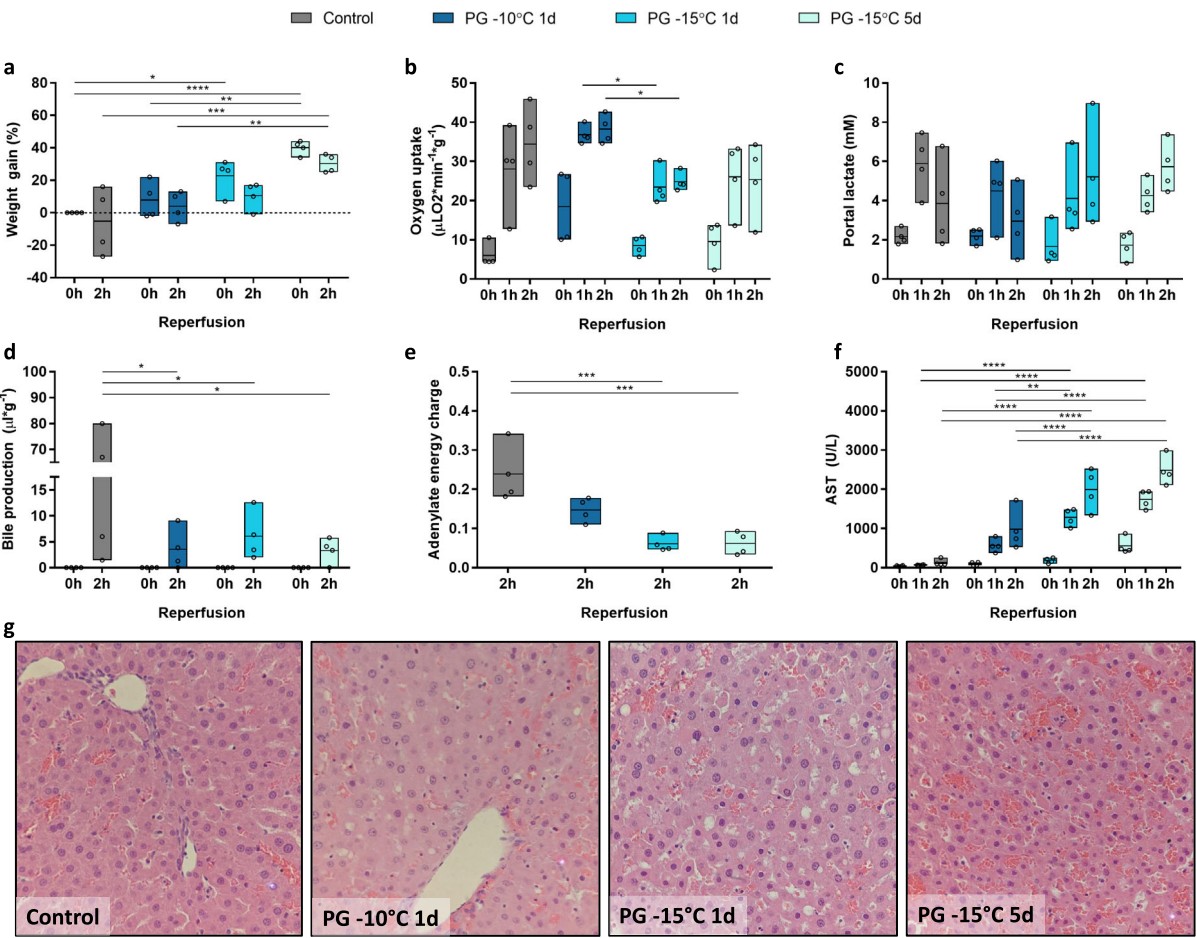

**Fig. 3 Effect of storage temperature (−10 vs. −15 °C) and duration of storage (1 vs. 5 days) on liver function, injury, and microscopic tissue structure after partial freezing with propylene glycol. a** Weight gain as percentage of the procurement weight. **b** Oxygen uptake. **c** Lactate concentration in the portal vein. **d** Bile production. **e** Tissue adenylate energy charge. **f** Aspartate aminotransferase concentration (AST) in the IVC. **g** Light microscopy images of parenchymal liver wedges at the end of simulated transplantation (×40). Controls (gray) = 1 day hypothermic preservation, PG = propylene glycol stored for 1 day at −10 °C (dark blue), 1 day at −15 °C (blue), and 5 days at −15 °C (light blue). Stars denote statistical significance (two-way ANOVA, followed by Tukey's post-hoc test): *0.01 < $p$ < 0.05; **0.001 < $p$ < 0.01; ***0.0001 < $p$ < 0.001; ****$p$ < 0.0001. Boxes: Floating bars (min to max), with a line at the mean. Source data are provided as a Source Data file.

At the end of simulated transplantation, potassium levels were significantly elevated in experimental groups (Clinical Oxy. PG −15 1 day was 5.23 ± 0.252 vs 5 day was 5.167 ± 0.404 mM) as compared to Clinical Oxy. Controls (3.65 ± 0.311; Fig. 5a). While the maximum AST (Fig. 5b) and ALT (Fig. 5c) values for partially frozen livers at −15 °C for 1 (AST 958.67 ± 278.50 U/L; ALT 722.00 ± 155.74 U/L) and 5 day (AST 1190.33 ± 117.35 U/L; ALT 886.67 ± 241.80 U/L) were significantly elevated above controls (AST 49.00 ± 4.00 U/L; ALT 23.33 ± 8.39 U/L), these values are nonetheless favorable in comparison to other literature values indicative of viable livers (i.e. ~3000 U/L)[46,47]. We also quantified changes in DAMPs, including cfDNA (Supplementary Fig. S8a), 8-OHdG (Supplementary Fig. S8b), TNFα (Supplementary Fig. S8c), and HSP70 (Supplementary Fig. S8d). While an increasing trend was observed in cfDNA and HSP70 collected from partially frozen versus control livers, this mostly did not reach statistical significance, except for an increase in HSP70 between 1 day frozen and Clinical Oxy. Controls. In contrast, there were no changes in TNFα levels across all groups and 8-OHdG values were significantly lower in frozen livers versus controls. Further, we assessed liver viability markers, including Albumin (Supplementary Fig. S9a), LDH (Supplementary Fig. S9b), FMN (Supplementary Fig. S9c), IL8 (Supplementary

Fig. S9d), NADH/NAD ratio (Supplementary Fig. S9e), and NADPH/NADP ratio (Supplementary Fig. S9f) that have been shown to be indicative of viability, metabolic perturbations, and/or function[12,48–53]. There were no statistically significant differences between controls and partially frozen livers for all markers, except LDH (Supplementary Fig. S9b) whereby values after 5 days in the frozen state were significantly elevated above all other groups. Finally, analysis of microscopic tissue structure (Fig. 4e) showed livers stored at −15 °C for 1 day showed intact portal triads, with central veins showing congestion. Livers stored for 5 days also showed focal endothelial disruption and sinusoidal congestion.

In addition to a 2-h simulated transplantation, we also present perfusion outcomes during a 6-h simulated transplantation in Table 1 and Supplementary Fig. S10. For this 6-h simulated transplantation, we compare control data (1-day hypothermic preservation with the Affinity Pixie Oxygenator, denoted as "Clinical Oxy. Control") versus livers that were partially frozen in the presence of propylene glycol (PG) at −15 °C for 5 days with the Affinity Pixie Oxygenator (denoted as "Clinical Oxy. PG −15 °C 5d"). While perfusion-based criteria to define liver viability can vary across countries/institutions, the VITTAL (viability testing and transplantation of marginal livers) clinical

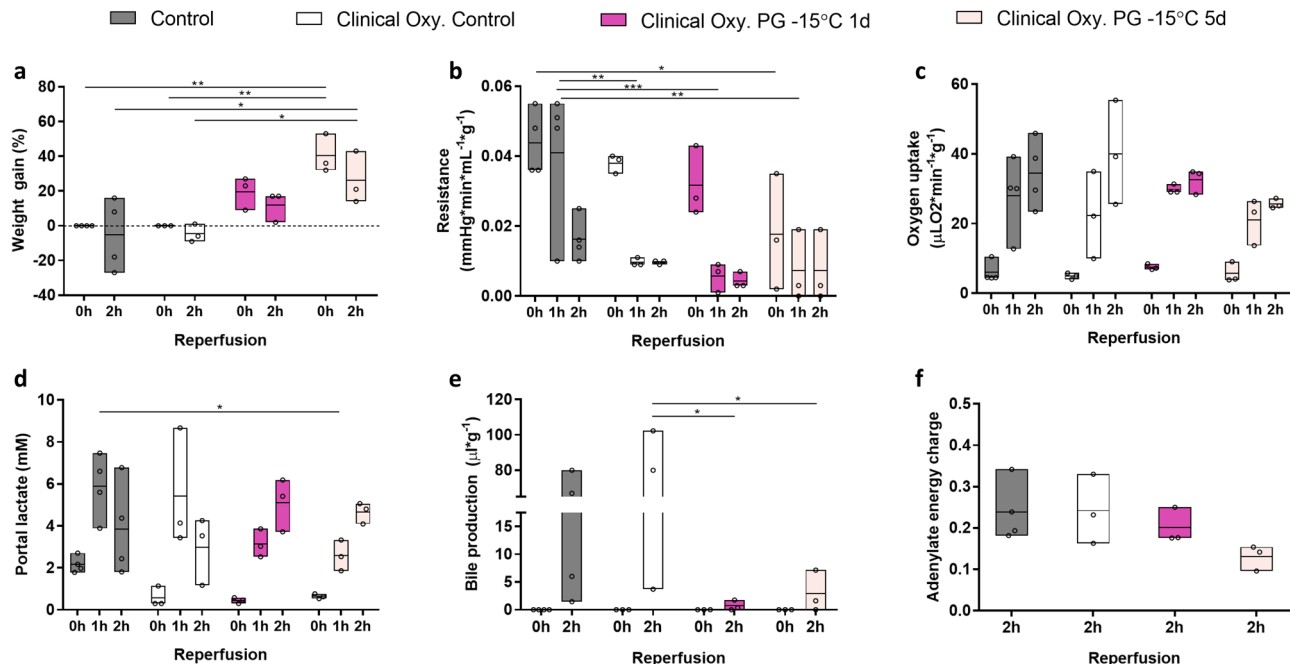

**Fig. 4 Effect of a clinical grade oxygenator on the recovery of liver function after partial freezing with propylene glycol at −15 °C for 1 and 5 days.**
**a** Weight gain as percentage of the procurement weight. **b** Vascular resistance between the portal vein and the IVC. **c** Oxygen uptake. **d** Lactate concentration in the portal vein. **e** Bile production. **f** Tissue adenylate energy charge. Control (gray) = 1 day hypothermic preservation, Clinical Oxy. Control (white) = 1 day hypothermic preservation perfused using a clinical grade oxygenator, PG = propylene glycol stored for 1 day at −15 °C (dark pink) or 5 days at −15 °C (light pink) perfused using a clinical grade oxygenator. Stars denote statistical significance (two-way ANOVA, followed by Tukey's post-hoc test): *$0.01 < p < 0.05$; **$0.001 < p < 0.01$; ***$0.0001 < p < 0.001$; ****$p < 0.0001$. Boxes: Floating bars (min to max), with a line at the mean. Source data are provided as a Source Data file.

trial defines livers as suitable for transplantation when the following criteria are met: organs metabolizing lactate to ≤2.5 mmol/L within 4 h of the perfusion commencing in combination with two or more of the following parameters—bile production, metabolism of glucose, a hepatic arterial flow rate ≥150 mL/min and a portal venous flow rate ≥500 mL/min, a pH ≥7.30 and/or maintain a homogeneous perfusion[54].

As presented in Table 1 and Supplementary Fig. S10, all livers metabolized lactate to levels that were not significantly different with controls at 6 h (Supplementary Fig. S10a; 2 of 3 livers reached values ≤2.5 mmol/L within 4 h of the perfusion), all produced bile (Supplementary Fig. S10b; significantly lower than controls at 6 h), all metabolized glucose (Supplementary Fig. S10c; no significant difference with controls at 6 h), all reached standard portal flow rates used in rodent liver NMP (Supplementary Fig. S10e; no significant difference with controls at 6 h), all maintained homogeneous perfusion (Supplementary Fig. S10g), although pH declined as a function of perfusion (Supplementary Fig. S10d). However, it should be noted that our protocol uses whole blood for the complete 6-hour duration, as compared to other protocols that typically use packed RBCs. Finally, while mean AST and ALT values after 6 h perfusion for partially frozen livers at −15 °C for 5 days (Supplementary Fig. S10f and Table 1; AST 1270 ± 430 U/L; ALT 767.7 ± 205.9 U/L) were significantly elevated above controls (AST 79.3 ± 33.5 U/L; ALT 34.7 ± 19.5 U/L), these values are nonetheless favorable in comparison to other literature values indicative of viable livers (i.e. <~3000 U/L)[46,47,55].

## Discussion

The donor organ shortage indirectly claims hundreds of thousands of lives each year[1]. The limited preservation time of donor organs has repeatedly been identified as a key bottleneck in organ transplantation and additionally hinders translation of emerging technologies such as immune tolerance induction and organ engineering approaches[1,8,19,42]. While benefits of extremely low preservation temperatures that near-completely halt metabolism and theoretically enable lifetime organ banking are obvious, both deep cryogenic freezing and vitrification approaches have inherent limitations that are directly coupled to the storage temperatures and have proven hard to overcome[17,18,42]. Instead of lifetime banking of donor organs, the current need for extended preservation can be met by extending the preservation time from a matter of hours to several days or weeks which does not necessitate such low storage temperatures. In this capacity, we explored liver preservation at high-subzero temperatures (−10 °C to −15 °C) to benefit from metabolic rate depression while abating challenges in deep cryogenic preservation. Inspired by nature, literature, and our previous experience with supercooling preservation, we developed a protocol for gradual CPA delivery/removal, reducing ice-dependent injury during high subzero freezing of rodent livers, and improving our machine perfusion system for optimal organ recovery/assessment.

Preservation endeavors of whole organs in the frozen state at high subzero temperatures are limited and have thus far been unsuccessful. With the first report dating back to 1966, canine livers were frozen for 1 day to 2 weeks at −20 °C using Glycerol (33 % vol/vol). Substantial endothelial injury and disruption of tissue architecture was observed, and no animals survived after orthotopic transplantation[23]. Three decades later, attempts were made to translate learnings from freeze-tolerant animals to mammalian livers. Rat livers were frozen at −2 °C for 2 h using glycerol (GLY) and antifreeze proteins derived from arctic fish. Despite relatively high subzero temperatures and short storage durations, success was limited by excessive endothelial injury after freezing[21,22]. Learning from these first attempts more than

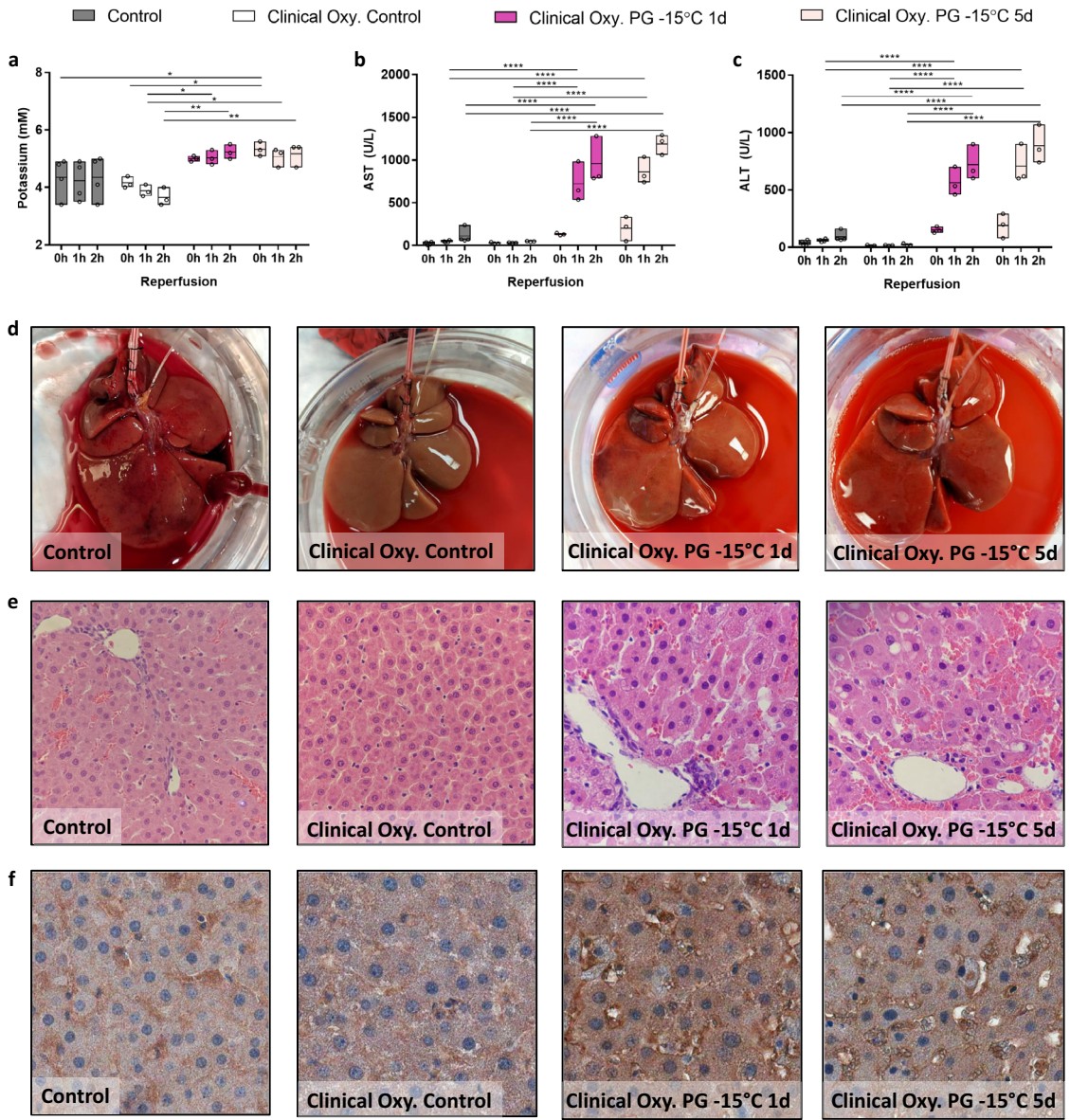

**Fig. 5 Effect of a clinical grade oxygenator on hepatocellular injury and microscopic tissue structure after partial freezing with propylene glycol at −15 °C for 1 and 5 days. a** Potassium concentration in the IVC. **b** Aspartate aminotransferase concentration (AST) in the IVC. **c** Alanine aminotransferase concentration (ALT) in the IVC. **d** Photos of the livers at the end of simulated transplantation. **e** Light microscopy images of parenchymal liver wedges at the end of simulated transplantation (×40). **f** Immunohistochemistry light microscopy images of Glut2 (a hepatocyte marker) at the end of simulated transplantation (×40). Control (gray) = 1 day hypothermic preservation, Clinical Oxy. Control (white) = 1 day hypothermic preservation perfused using a clinical grade oxygenator, PG = propylene glycol stored for 1 day at −15 °C (dark pink) or 5 days at −15 °C (light pink) perfused using a clinical grade oxygenator. Stars denote statistical significance (two-way ANOVA, followed by Tukey's post-hoc test): *$0.01 < p < 0.05$; **$0.001 < p < 0.01$; ***$0.0001 < p < 0.001$; ****$p < 0.0001$. Boxes: Floating bars (min to max), with a line at the mean. Source data are provided as a Source Data file.

20 years ago, we aimed to mimic freeze-tolerant strategies in nature and show functional recovery of partially frozen grafts. In recognizing the challenges of high subzero freezing approaches, especially with respect to endothelial injury, we recently leveraged tissue engineering techniques to build on current knowledge and interrogate the impact of cryopreservation approaches on endothelial cells adherent to a structural scaffold[20]. With direct microscopic observation in endothelialized microchannels, we introduced an INA (Snomax) and intracellular and extracellular CPAs (3-O-methly-D-glucose and polyethylene glycol, respectively) to rescue endothelial cells from injury during partial freezing at −10 °C, thereby overcoming a significant barrier in previous literature.

With a deeper understanding of endothelial injury during high subzero freezing and identification of key protective strategies, we focused our initial optimization strategies on safely delivering CPAs to protect endothelial cells. Others have also shown gradual and controlled perfusion of cryoprotective agents is required to avoid injury[56–60]. We built on this previous work in the present protocol by using gradual (un)loading and the presence of osmotic counter ions. Next, we compared the effects of glycerol (GLY), ethylene glycol (EG) and propylene glycol (PG) on liver viability after partial freezing at −10 °C to a control group of conventional hypothermic preservation (+4 °C), both of which were stored for 24 h. We showed that the use of PG as the main permeating CPA during partial freezing improves liver function and reduces injury compared to the use of GLY or EG

**Table 1 Rodent livers that were partially frozen at −15 °C for 5 days are viable using our 6-our perfusion model as compared to transplantable controls and/or clinical liver perfusion data, representing a fivefold extension of preservation duration.**

| Viability index | Partial freezing @ −15 °C for 5 days | Hypothermic control @ 4 °C for 1 day | Clinical EVLP |
|---|---|---|---|
| Lactate | All livers metabolized lactate. Mean 2.65 ± 0.77 mM at 4 h and 1.843 ± 0.97 mM at 6 h. | No difference with frozen livers. | Organs metabolizing lactate to ≤2.5 mmol/L at 4 h[54]. |
| Bile production | All livers produced bile. | Total accumulation was higher than frozen livers. | Livers are producing bile[54]. |
| Glucose | All livers metabolized glucose. | No difference with frozen livers. | Metabolism of glucose[54]. |
| pH | pH decreased throughout perfusion. | pH was higher than frozen livers. | A pH ≥ 7.30[54]. |
| Flow/Resistance | All livers reached target flows. | No difference with frozen livers. | A hepatic arterial flow rate ≥150 mL/min and a portal venous flow rate ≥500 mL/min[54]. |
| Uniformity | Homogeneous perfusion achieved. | No difference with frozen livers. | Maintain a homogeneous perfusion[54]. |
| Injury markers | | | |
| ALT | Mean 768 ± 206 U/L. | ALT lower than frozen livers. | ALT ≤ ~3000 U/L[46]. Range 152–1460 U/L[55]. |
| AST | Mean 1270 ± 430 U/L. | AST lower than frozen livers. | Range 227–9200 U/L[55]. |

Controls are livers that were stored at 4 °C in the University of Wisconsin solution for 24 h. Perfusion-based metrics that are used in clinical ex vivo liver perfusion (EVLP) to define liver suitability for transplantation include lactate falling below 2.5 mM in combination with 2 out of 5 other indicators (bile production, glucose metabolism, venous/arterial flow rates, pH, and uniformity of perfusion)[54]. Rodent liver data and statistical results presented are from the final, 6-h time point of our simulated transplantation model, unless otherwise stated.

(Figs. 2 and S2). Importantly, liver function – based on oxygen uptake and lactate clearance – in livers that were frozen for 24 h at −10 °C with PG was the same as transplantable control livers. Impaired oxygen consumption has been shown to correlate to transplant survival in preclinical studies[13,61,62]. Whereas the implications of oxygen consumption during NMP in clinical setting are unclear, lactate clearance translated well to clinical studies and is considered a parameter of liver viability[2,41,54,63,64]. In contrast to lactate and oxygen consumption, all frozen livers produced significantly less bile and released significant more transaminases after freezing, as compared to controls. Preclinical studies showed that reduced bile production and elevated transaminase levels negatively correlate to transplant survival[13,61]. In the clinical setting, bile production and low transaminase levels during NMP are considered favorable however not necessary for safe transplantation of the graft if other parameters are viable[41,54,65]. Nonetheless, in the present study we consider these as important signs of hepatocellular injury after freezing that need to be addressed.

The significant differences in post freezing viability as a function of various permeating CPAs could be explained by several factors. First, GLY, EG, and PG have different toxicity profiles. These agents were loaded at hypothermic temperatures to reduce potential toxicity; however, even at low temperatures potential toxicity could be a factor. Although specific information about liver toxicity of the used CPAs at the temperatures and concentrations we implemented is limited, in general PG is considered less toxic than EG which is consistent with the post freezing perfusion metrics in the present study[17]. Second, the CPAs have different membrane permeabilities, with EG being the fastest and GLY the slowest to diffuse through cell membranes. To reduce the osmotic gradients, we gradually increased the CPA concentrations and used the extracellular (non-permeating) CPAs trehalose and raffinose. However, the increase in CPA concentration during preconditioning and the trehalose and raffinose concentrations were the same between groups. This may have caused differences in transmembrane osmotic stress between the experimental groups. Also, the CPAs have different physical properties that influence ice formation – witnessed by the differences in CPA-water phase diagrams[65,66]—which can impact freezing injury. Finally, it should be noted that equivalent volumes of these permeating CPAs were used in the final storage solution (see Supplementary Table S1), yet their molecular masses differ (EG 62.07 g/mol, PG 76.09 g/mol, GLY 92.09 g/mol). Since

depression of freezing point or amount of total ice at a given temperature is a molar effect, these differences could be one factor that influenced outcomes.

With the choice of carrying forward PG as the main permeating CPA, we aimed to systematically address the effect of deeper storage temperatures (by comparing −10 and −15 °C storage temperatures for 24 h) vs longer preservation times (−15 °C storage for 24 h vs 5 days; Figs. 3 and S3). First, the reduction in temperature from −10 °C to −15 °C resulted in significantly reduced metabolic function after partial freezing. This was evident from a reduction in oxygen uptake as well as lack of lactate clearance during simulated transplantation in the livers that were partially frozen at −15 °C. In addition to the negative impact on liver function, the reduction of the freezing temperature also led to a significant increase in hepatocellular and endothelial injury, as evidenced by higher levels of transaminase release during simulated transplantation and loss of LSECs as observed through histological analysis.

Lowering the freezing temperature has three potential consequences that may explain the observed reduction in liver viability at −15 °C as compared to −10 °C. First, when water freezes, solutes are excluded from the growing ice crystal which increases the osmolarity and therefore decreases the freezing point of the unfrozen water fraction. This results in a physical equilibrium between the temperature and the amount of ice present in the organ. Thus, a lower freezing temperature results in more ice and osmotic shock, resulting in potentially more injury. Second, the physical structure and dynamics of the ice crystal growth is directly dependent on temperature[67] and becomes increasingly injurious at lower temperatures. Although experimental studies and natural model systems suggest this is especially the case at lower temperatures than those used in the present study, this effect cannot be completely excluded. Third, cold temperatures are known to increase the rigidity of lipid membranes. For supercooling preservation, we overcame this by using 35 kDa PEG to provide cell membrane stabilization at subzero temperatures[14]. We anticipated this effect may be aggravated at the even lower temperatures during partial freezing. Therefore, we leveraged the cell stabilizing properties of trehalose and raffinose[68] in addition to PEG during partial freezing at −10 °C. However, it must be noted that the PEG, trehalose, and raffinose concentrations were the same in both −10 °C and −15 °C experimental conditions.

We also aimed to evaluate the impact of longer preservation durations by comparing 24 h vs 5 day preservation at −15 °C (Figs. 3 and S3). These preservation lengths were chosen since for every 10 °C temperature reduction, the metabolic rate is approximately reduced by a factor 2–2.5[14]. Further, based on the 100% transplant successes that our group previously achieved after 24 h storage at 4 °C and 72 h supercooled storage at −6 °C followed by SNMP recovery, we estimated that a storage temperature of −15 °C with SNMP recovery would enable an increase in the preservation duration to 5 days[14,15,61]. Increasing the storage duration from 1 to 5 days caused a moderate increase in edema while all other viability parameters of liver function and injury during simulated transplantation were largely unaffected by the substantially increased storage duration. In conjunction with the results of the 24 h partially frozen livers at −15 °C, this suggests that the major contributor to injury is the process of freezing/thawing and not extended subzero preservation duration. It seems that the low storage temperature provides adequate depression of metabolism and livers can be stored for 5 days without considerable reduction in viability as compared to livers stored for 1 day at the same temperature. However, we emphasize that this deduction should be made with caution as both the 1- and 5-day frozen livers show reduction in viability after partial freezing at −15 °C.

The flexibility, accessibility, and low cost of non-clinical grade oxygenators enabled careful characterization of the effects of permeating CPAs as well as storage temperature and duration on liver viability after partial freezing. While some functional parameters, such as vascular resistance, were favorable during simulated transplantation, livers nonetheless showed evidence of injury after freezing. In particular, tissue adenylate energy charge is considered one of the representative metrics for liver viability[35,42–45], and low transaminase levels during NMP are considered favorable (although not necessary in of itself) for safe transplantation[41,54,65]. These two parameters showed unfavorable results as compared to control for livers partially frozen at −15 °C for both 1 and 5 days (Figs. 3 and S3). To overcome this issue, we incorporated a clinical grade oxygenator that has graduated fiber bundle density technology and radial flow path that is designed to create a more uniform blood flow, enhance gas transfer, and reduce pressure drops. Moreover, it also contains a hydrophilic polymer coating to reduce platelet adhesion and activation, which would reduce coagulation and inflammatory events that may be triggered through interaction of recirculating whole blood with the perfusion system. As a result, despite a 5-fold extension of preservation duration, tissue adenylate energy charge was not statistically different from controls after a 2-hour simulated transplantation and the maximum AST/ALT values were favorable after 2-h and 6-h simulated transplantation in comparison to other literature values indicative of viable livers[46,47] (Figs. 4f, 5b, c, S10f and Table 1). Further, after a 6-h simulated transplantation, all livers that were stored for 5 days at −15 °C showed endpoint values that can be described as either of the following: (i) were not significantly different from controls or (ii) considered acceptable based on clinical criteria[46,54,55]. The only exceptions were pH, which decreased as a function of perfusion for the 5-day frozen livers (Supplementary Fig. S10 and Table 1), and it should be noted that low cumulative bile production and elevated transaminase levels remain a limitation of the study.

In summary, we developed a protocol for partial freezing of rat livers for the longest storage duration (up to 5 days) and deepest high subzero storage temperatures (−10 °C to −15 °C) that incorporates advances in the field of cryopreservation and leverages technologies from the emerging field of machine perfusion. Using this protocol, we demonstrated multiple strategies that improve post freezing liver viability, including the process of (un)loading, strategic selection of CPAs, and addressing key engineering aspects of machine perfusion. Essential next steps are to confirm long-term viability with an established transplant model in rodent and swine.

## Methods

**Ethical statement**. All research complies with all relevant ethical regulations and the experimental protocol was approved by the Institutional Animal Care and Use Committee (IACUC) of Massachusetts General Hospital (Boston, MA, USA; 2017N000227).

**Experimental design**. As shown in Fig. 1, the liver partial freezing protocol entails 9 consecutive steps; (1) procurement of the liver (2) preconditioning during sub-normothermic machine perfusion (SNMP), (3) preloading of CPAs during hypothermic machine perfusion (HMP), (4) loading of the final storage solution during HMP, (5) freezing, (6) thawing, (7) unloading of CPAs during HMP, (8) functional recovery during SNMP, and (9) viability assessment during ex vivo simulated transplantation.

Within this standardized protocol we first compared GLY, EG, and PG as main permeating CPA using a freezing temperature of −10 °C and storage duration of 1 day (Fig. 2 and Supplementary Figs. S2, S4). Using the CPA that gave the best results (PG), we subsequently tested the effect of freezing temperature (−10 °C vs −15 °C) and storage duration (1 day vs 5 days) on liver viability after partial freezing using the same protocol (Fig. 3 and Supplementary Figs. S3, S5). Once we had a good understanding of the effects of various permeating CPAs as well as the effect of storage temperature and duration on liver viability after freezing, we improved our perfusion system by substituting the Radnoti (Cat# 130144) with an Affinity Pixie Oxygenator (Cat# BBP241; Figs. 4–5 and Supplementary Figs. S6, S8). The Affinity Pixie Oxygenator is a clinical grade oxygenator that has graduated fiber bundle density technology and radial flow path that is designed to create a more uniform blood flow, enhance gas transfer, reduce pressure drops, and decrease prime volume. Moreover, it also contains a Balance® Biosurface that contains a hydrophilic polymer coating to reduce platelet adhesion and activation.

An additional control group of 1 day HP (+4 °C) was included as our previous studies have repeatedly shown this duration of conventional preservation results in 100% transplant survival in the same animal model. In addition to this control group, we also cite relevant outcome measurements as compared to data presented herein to provide additional context. After freezing, liver viability during SNMP recovery and ex vivo simulated transplantation was compared between the experimental group and a hypothermic preservation control group.

**Machine perfusion system**. The perfusion system provides a continuous perfusion through the portal vein that is pressure, flow, and temperature controlled. The perfusates can be either recirculated or flushed with a single pass though the liver and perfusates can be changed without interrupting perfusion. The setup and operation of the perfusion system is described in detail elsewhere[15]. There were two oxygenators used in this study, including the Radnoti (Cat# 130144) and Affinity Pixie Oxygenator (Cat# BBP241). Due to the flexibility and low cost of the Radnoti oxygenator, it was first used for initial optimization and characterization studies, including determining the effect of permeating CPAs (GLY vs EG vs PG; Fig. 2 and Supplementary Figs. S2, S4), temperature (−10 vs −15 °C, Fig. 3 and Supplementary Figs. S3, S5) storage duration (1 vs 5 days, Fig. 3 and Supplementary Figs. S3, S5), and warm ischemic injury (Supplementary Fig. S7). Once the major aspects of the protocol were optimized and characterized, we used the clinical grade Affinity Pixie Oxygenator in Figs. 4, 5 and Supplementary Figs. S6, S8.

**Liver procurement**. The experimental protocol was approved by the Institutional Animal Care and Use Committee (IACUC) of Massachusetts General Hospital (Boston, MA, USA; 2017N000227). Lewis rats (strain code 004) were socially housed in temperature (70 °F ± 2 °F) and humidity (30–70%) controlled environments within pathogen-free HEPA filtered ventilated cages, with alternating 12-h light/dark cycles. Animals were provided sterilized standard rat chow and water ad libitum. Livers were procured from male rats (250–300 g; age 10–12 weeks; Charles River Laboratories, Wilmington, MA, USA). The rats' bile duct was cannulated and were then heparinized with 30U (see Supplementary Table S2 for suppliers of reagents). The portal vein's splenic, gastric branches and hepatic artery were all ligated. The portal vein was then cannulated with a 16-gauge catheter and immediately flushed with 40 ml heparinized saline (1000 U/ml at room temperature). Next the liver was freed from the abdomen and flushed with and additional 20 ml of heparinized saline to remove any residual blood within the liver. The perfusion was always initiated within 5 min of procurement, except for livers that were exposed to a warm ischemic event. After procurement, warm ischemic livers were held at 34 °C for 30 min in the presence of lactated ringers solution prior to initiation of perfusion, as we have done before[62].

**Hypothermic preservation**. During procurement the livers were flushed with ice cold instead of room temperature heparinized saline. After removal of the liver from the abdomen, the livers were directly flushed with 30 ml of ice-cold University

of Wisconsin solution (UW). The livers were place in a bag with 50 ml of ice-cold UW and stored at 4 °C for 24 h.

**Partial freezing protocol**. Directly after procurement the livers were perfused with 250 ml of SNMP preloading solution (see Supplementary Table S1 for composition of all solutions). The perfusion temperature was set at 21 °C and perfusion was initiated at 5 ml/min. The flows were gradually increased (1 ml/min) until a maximum perfusion pressure of 5 mmHg or flow of 25 ml/min was reached. Next, the livers were perfused for 30 additional minutes to allow for cellular uptake of 3-O-methyl-d-glucose (3-OMG).

The SNMP preloading solution was gradually switched to HMP preloading solution in 10, 25 ml increments, each with 10% decreasing and 10% increasing volumetric fractions of SNMP and HMP preloading solution respectively. During this switch, the perfusion temperature was lowered to 4 °C and flows adjusted to a maximum perfusion pressure of 3 mmHg. Dependent on the flow rate, the solution switch took 10–15 min. The HMP preloading was continued for 30 additional minutes to ensure complete equilibration of the solution throughout the peripheral liver tissue.

Next, the CPA concentrations were gradually increased whereby the base solution was switched from William's Medium E (WE) to University of Wisconsin (UW) solution. This was done in 10 fractions of 10 ml, each with 10% decreasing and 10% increasing volumetric fractions of HMP preloading solution and storage solution, respectively. During this switch the perfusion temperature was maintained at 4 °C and the flow rates were lowered to ensure a maximum perfusion pressure of 3 mmHg. The final 15 ml fraction of 100% storage solution was loaded at a fixed flow rate of 0.5 ml/min.

After loading of the storage solution, the liver was placed in a bag with 50 ml fresh storage solution and suspended in a pre-cooled chiller (Engel, Schwertberg, Austria). Depending on the experimental condition the chiller temperature was pre-cooled to −10 °C or −15 °C and the liver stored for 1 of 5 days.

After storage, the liver with the frozen storage solution was removed from the bag and placed in a 37 °C bath (Thermo Fisher, Waltham, MA, USA) with 50 ml thawing solution. The bath was turned off and the liver was gently agitated until thawed. This took 5 min and resulted in a final bath and a liver surface temperature of 4 °C.

The liver was connected to the perfusion system and perfused with thawing solution for 30 min at a constant flow rate of 2 ml/min and a temperature of 4 °C. Next the perfusion temperature was increased to 21 °C and flows adjusted to a maximum perfusion pressure of 5 mmHg once the liver reached 21 °C. Also, the thawing solution was gradually switched to SNMP recovery solution in 10, 25 ml increments, each with 10% decreasing and 10% increasing volumetric fractions of thawing and SNMP recovery solution, respectively.

After rewarming and gradual removal of the storage solution, the livers were perfused with 300 ml of SNMP recovery solution that was recirculated during 3 h of perfusion using a maximum perfusion pressure of 5 mmHg and up to a flow of 25 ml/min.

**Simulated transplantation model**. All whole blood from one Lewis rat (~13 ml) was reconstituted up to a total volume of 100 ml with supplemented WE solution (Supplementary Table S1). The whole blood solution was stored at 21 °C and always used within 4 h of blood draw. The perfusion system was emptied and primed with 100 ml whole blood solution and the temperature was increased to 37 °C while the livers were briefly disconnected from the system to be weighed. For the control group, the system was primed the same and the UW solution was flushed from the livers with 25 ml of ice-cold saline before the livers were connected to the machine perfusion system. For all livers a maximum perfusion pressure of 11 mmHg and up to a flow of 30 ml/min was used once the livers reached a normothermic temperature of 37 °C and reperfusion was continued for 2 h and 6 h.

**Viability assessment**. Perfusate measurements were performed hourly during the functional recovery and ex vivo simulated transplantation, unless otherwise specified. PO2, O2 saturation, pH, and Lactate were measured in the inflow (portal vein) and outflow (infrahepatic vena cava), and potassium and glucose were measured only in the outflow using an i-STAT blood analyzer (Abbott, Chicago, IL, USA). AST and ALT concentrations were measured in the outflow using the Piccolo Xpress Chemistry Analyzer (Abbott). Perfusate samples were also collected at 1 h during simulated transplantation for quantification of damage-associated molecular patterns (DAMPs). Perfusate samples were centrifuged at $5000 \times g$, and the plasma stored at −80 °C for subsequent analysis. Enzyme-linked immunosorbent assays were performed according to manufacturer instructions for the following: cell-free DNA (Thermo Fisher, P7589), heat shock protein 70 (HSP70, Abcam, ab133061), tumor necrosis factor-alpha (TNFα, Abcam, ab236712), 8-hydroxy 2 deoxyguanosine (8-OHdG, Abcam, ab201734), Albumin (Abcam, ab108789), Lactate Dehydrogenase (LDH, MyBioSource, MBS2018912), Flavin Mononucleotide (FMN, MyBioSource MBS2510190), and Interleukin-8 (IL8, MyBioSource MBS9141543).

Liver weight was measured directly after procurement, just prior to freeze, post thaw, post recovery, and after simulated transplantation. Weight gain was calculated as the percentage weight increase and any time point compared to the liver weight after procurement.

Cumulative bile production was measured by weighing the bile-containing Eppendorf tube on a microscale at the end the functional recovery step of the partial freezing protocol and at the end of simulated transplantation.

Wedge biopsies were taken at the end of simulated transplantation. Biopsies were either flash frozen in liquid nitrogen and stored at −80 °C or fixed in buffered 5% formaldehyde for 24 h and stored in 70% ethanol. Flash-frozen biopsies were used to quantify adenosine triphosphate (ATP), adenosine diphosphate (ADP), adenosine monophosphate (AMP), nicotinamide adenine dinucleotide (NADH and NAD), and nicotinamide adenine dinucleotide phosphate (NADPH and NADP) by the Mass Spectrometry Special Shared Facilities at Shriners Hospitals for Children (Boston), as described elsewhere[35]. In short, the tissue was homogenized in liquid nitrogen and analyzed with targeted multiple reaction monitoring on a 3200 triple quadrupole liquid chromatography-mass spectrometry system (AB Sciex).

Fixed biopsies were processed (including embedded in paraffin) and hematoxylin and eosin staining was performed by the Massachusetts General Hospital Histology Molecular Pathology Core. Additionally, the IHC staining was performed using Dako EnVision®+ System-HRP (Dako, Carpinteria, CA, USA). The sections were de-paraffinized in an incubator at 58 °C for 1 h, then processed in xylene and rehydrated through a series of graded alcohols. Antigen retrieval was performed by incubating the sections in a citrate buffer (pH 6.0) for 20 min in a steamer. After antigen retrieval, the specimens were allowed to cool for 30 min, and endogenous peroxidase activity was blocked for 30 min in a solution containing 3% hydrogen peroxide. Slides were then incubated for 1 h at room temperature using a rocker, using GLUT-2 primary antibody (1:200, rabbit, polyclonal, Proteintech cat # 20436-1-AP). After incubation with the primary antibody, the slides were incubated with peroxidase labeled polymer for 30 min and developed with diaminobenzidine (DAB) (Biocare Medical, cat# BDB2004), followed by Mayer's hematoxylin counterstaining (Biocare Medical, cat# H-3404-100). The slides were then mounted with ProLong™ Glass Antifade Mount (Thermo, cat# P36980). Hematoxylin and eosin-stained slides and IHC slides were blindly assessed by an experienced liver pathologist (E.O.A.H). Processed slides were scanned under ×40 magnification using an Aperio ImageScope (Leica Biosystems).

Vascular resistance was calculated by dividing the perfusion pressure in the portal vein by the flow rate that was then corrected for weight of the liver after procurement. The oxygen uptake rate (OUR) was calculated with the following formula: $(aO_2*(Port\_pO_2*port\_flow - Ven\_pO_2*port\_flow) + Hb*cHb*(Port\_sO_2/100*Port\_flow - Ven\_sO_2/100*Port\_flow))/Liver\_weight$. Where $aO_2$ is oxygen solubility coefficient ($3.14 \times 10^{-2}$ µLO2 per mmHgO2 per ml); Port_pO2 is portal partial oxygen pressure (mmHg); Ven_pO2 is venous partial oxygen pressure (mmHg); Port_flow is portal flow rate (ml min−1); Port_sO2 is portal hemoglobin saturation (%); Ven_sO2 is venous hemoglobin saturation (%); cHb is hemoglobin oxygen-binding capacity (1.34 ml O2 g−1); Hb is hemoglobin (g ml−1); and Liver_weight is liver weight (g), as others have done[13]. Energy charge was calculated with the following formula: $ATP + 0.5ADP/(ATP + ADP + AMP)$.

**Statistical analysis**. All data was compiled Microsoft Excel 365. All graphing and statistical analyses were performed with Prism 7.03 (GraphPad Software Inc., La Jolla, CA, USA) with a (two-sided) significance level of 0.05. Repeated measures two-way ANOVAs were used for the comparison of the time-course perfusion data, followed by Tukey's post-hoc test to examine statistical differences between the experimental groups and to correct for multiple comparisons.

**Reporting summary**. Further information on research design is available in the Nature Research Reporting Summary linked to this article.

## Data availability

The authors declare that the data supporting the findings of this study are available within the paper and its Supplementary information files. Source data are provided with this paper.

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

## Acknowledgements

This research was funded from the US National Institutes of Health, including R01DK114506 (M.T., K.U., M.L.Y.), R01DK096075 (K.U., H.Y.), R01DK107875 (K.U.), R01EB028782 (KU), and National Science Foundation under Grant No. EEC 1941543 (M.T.). Further, we gratefully acknowledge funding to SNT from NIH (K99/R00 HL1431149; R01HL157803), American Heart Association (18CDA34110049), Harvard Medical School Eleanor and Miles Shore Fellowship, and the Claflin Distinguished Scholar Award on behalf of the MGH Executive Committee on Research. We thankfully acknowledge support provided by the Tosteson Fellowship awarded to R.J.V. by the Executive Committee on Research at the Massachusetts General Hospital. We thank MGH Cell Resource Core, including Peony D. Banik and Sonal Nagpal, for assistance with perfusion equipment and the Mass Spectrometry Special Shared Facilities at Shriners Hospitals for Children (Boston) for quantification of tissue adenylate levels. Finally, we are grateful for our collaboration on high subzero organ preservation with Sylvatica Biotech, Inc and acknowledge awards R44 DK124053-01 (US NIH), W81XWH21C0060 (US Dept. of Defense) and support from ATCC.

## Author contributions

S.N.T., R.J.V., C.A.P., S.E.J.C., K.U., and M.T. conceived and designed the study; S.N.T., R.J.V., C.A.P., S.E.J.C., S.O., S.R., B.T.W., and J.P.O.C., performed the experiments; S.N.T., R.J.V., E.O.A.H., O.B.U., S.L.S., H.Y., M.Y., K.U., and M.T. analyzed and interpreted data; S.N.T. and R.J.V. performed statistical analysis and wrote the manuscript; S.N.T., R.J.V., C.A.P., S.E.J.C., S.O., T.M.G., O.B.U., S.L.S., H.Y., M.Y., K.U., and M.T participated in critical revision of the manuscript for intellectual content; All authors contributed to the preparation of the manuscript.

## Competing interests

The authors declare competing interests. M.T., K.U., R.J.V., S.N.T., O.B.U., S.L.S., M.L.Y., S.E.J.C., and C.A.P. have patent applications relevant to this study. These include: (1) ML Yarmush, TA Berendsen, R Bieganski, ML Izamis, S Perk, M Toner, OB Usta, B Uygun, MK Uygun (2011) Methods and Compositions for Preserving Tissues and Organs. US Patent Application No. 13/695,459 (Issued); US Patent Divisional Application No. 16/705,966 (pending). This invention relates to methods for high subzero non-freezing storage (i.e. supercooling). (2) SL Stott, SN Tessier, L Weng, M Toner (2016) Ice nucleation formulation for cryopreservation and stabilization of biologics. US Patent Application No. 16/313,714 (pending); Canada Patent Application No. 3,029.669 (pending); European Patent Application No. 17821252.8 (pending); Japan Patent Application No. 2018-568442 (pending). This invention relates to methods to initiate ice nucleation. (3) SEJ Cronin, RJ de Vries, CA Pendexter, SL Stott, SN Tessier, M Toner, MK Uygun, L Weng (2017) High subzero cryopreservation. US Patent Application No. 16/622,457 (pending); European Patent Application No. 18816781.1 (pending). This invention relates to methods for high subzero freezing storage. Further, KU has a financial interest in Organ Solutions, a company focused on developing organ preservation technology. M.T., K.U., and S.N.T. serve on the Scientific Advisory Board for Sylvatica Biotech Inc., a company focused on developing high subzero organ preservation technology. All competing interests are managed by the MGH and Mass General Brigham in accordance with their conflict-of-interest policies. The remaining authors declare no competing interests.
