## [Peer Review File · Nature Communications]

Reviewers' Comments:

Reviewer #1:

None

Reviewer #2:

Remarks to the Author:

This is an interesting study by Tessier and colleagues on partial freezing of rat livers towards a method for longer term storage of organs for transplantation. The authors chose to study high subzero temperatures of about -10 to -15 C which is the range where freeze tolerant species such as wood frog over winter. To achieve freeze tolerance a complex perfusion for gradual introduction of cryoprotectants was developed. Early results on functional analyses after partial freezing were encouraging but more work will be needed in future.

The paper is well presented and Figures are informative and clear.

There are a couple of small points which the authors might like to address.

1. The authors have developed a perfusion method for gradual increase of cryoprotectants into the liver which avoided osmotic and other damage. Their system is well described. They should just acknowledge that other authors also found in earlier work that gradual controlled perfusion of cryoprotectants was required to avoid injury. See work of David Pegg in 1970s 1980s. Although this was work in kidneys, the principles of perfusion were similar. The authors should think to add some discussion about these.

1.

Introduction and removal of cryoprotective agents with rabbit kidneys: assessment by transplantation.

Jacobsen IA, Pegg DE, Starklint H, Hunt CJ, Barfort P, Diaper MP.

Cryobiology. 1988 Aug;25(4):285-99. doi: 10.1016/0011-2240(88)90037-5.

PMID: 3136972

2.

Perfusion of rabbit kidneys with cryoprotective agents.

Pegg DE.

Cryobiology. 1972 Oct;9(5):411-9. doi: 10.1016/0011-2240(72)90158-7.

PMID: 4568154 No abstract available.

3.

Effect of cooling and warming rate on glycerolized rabbit kidneys.

Jacobsen IA, Pegg DE, Starklint H, Chemnitz J, Hunt C, Barfort P, Diaper MP.

Cryobiology. 1984 Dec;21(6):637-53. doi: 10.1016/0011-2240(84)90223-2.

PMID: 6394215

4.

Flow distribution and cryoprotectant concentration in rabbit kidneys perfused with glycerol solutions.

Pegg DE, Robinson SM.

Cryobiology. 1978 Dec;15(6):609-17. doi: 10.1016/0011-2240(78)90085-8.

PMID: 743886 No abstract available.

5.

Perfusion of rabbit kidneys with glycerol solutions at 5 degrees C.

Pegg DE, Wusteman MC.

Cryobiology. 1977 Apr;14(2):168-78. doi: 10.1016/0011-2240(77)90137-7.

PMID: 862413 No abstract available.

2. The authors chose to use equivalent volumes of ethylene or propylene glycol e.g. 60 ml in their perfusate formulations. The impact of solutes on depression of freezing point or amount of ice formed at a given temperature is a molar effect of depression of freezing. The molecular masses of EG and PDIol differ by about 15% - so the concentrations of active solutes will be different in their

perfusates. The authors might like to Discuss this to give readers a better understanding of the biophysics

Reviewer #3:

Remarks to the Author:

Summary

This is an experimental study on freezing of rat livers at subzero temperature, with the intention to extend preservation time. A combination of different machine perfusion techniques and storage was applied, and livers were tested during a formal ex vivo reperfusion period for 2 hours with full blood at normothermic temperature, to simulate a transplant procedure. Endpoints included weight gain, vascular resistance, portal flow, oxygen uptake, lactate in portal vein, bile production, potassium in the intrahepatic cava, perfusate AST; and ALT.

The authors compared in a first step effectiveness of three cytoprotective agents, such as glycerol, ethylene glycol and propylene glycol during a 24 hour preservation at -10 °C, compared to conventional cold storage. In a next step, a lower storage temperature at -15°C was tested. In a third step, the extension of the preservation to 5 days was compared to the 1 day results.

Criticisms

The manuscript is well written and the topic is timely. The authors should however better explain why they perform this study, as their previously published supercooling technique has already shown a clear extension of the preservation time to 4 days in rodents, and to 27h in human. What would be the benefit of exploring even deeper temperatures? Next, I would disagree that all transplantable organs need currently a clear extension of the preservation time. What is currently more needed in many countries, is a repair strategy of injured organs, and a reliable assessment of the degree of injury before use, which however both appears limited by a static approach, with deep metabolic depression during any freezing procedure. This should be further discussed.

I have several additional remarks and questions:

1. Why did the authors choose a model of healthy livers preserved for 24 hours? As stated, these livers are regarded to result in 100 % survival after transplantation. The applied freezing methods however did even under such conditions not reach the values of controls in every group, which is a clear hurdle, please comment on this.
2. Reperfusion on the isolated perfusion model is of limited value, especially if run only for 2 hours. To safely evaluate liver viability and function, much more endpoints would be needed, for example assessment of mitochondrial metabolites and energy charge, endothelial integrity, non-parenchymal cell activation, DAMP signaling, etc. In addition, I would also recommend histology staining for each type of liver cells.
3. All experiments should then be repeated in injured livers, for example in livers exposed to additional warm ischemia or steatotic livers.
4. The introduction and the discussion should consider previous results regarding supercooling, and explain why further research with deeper temperature is needed.
5. Finally, a transplant model should be used in all groups to convince on viability of frozen livers.

RESPONSE TO REVIEWERS

Reviewer #2: We thank the reviewer for careful consideration of the manuscript and thoughtful comments that have improved the manuscript. We provide a point-by-point response to all comments below:

Question 1: The authors have developed a perfusion method for gradual increase of cryoprotectants into the liver which avoided osmotic and other damage. Their system is well described. They should just acknowledge that other authors also found in earlier work that gradual controlled perfusion of cryoprotectants was required to avoid injury.

See work of David Pegg in 1970s 1980s. Although this was work in kidneys, the principles of perfusion were similar. The authors should think to add some discussion about these.

1. Introduction and removal of cryoprotective agents with rabbit kidneys: assessment by transplantation. Jacobsen IA, Pegg DE, Starklint H, Hunt CJ, Barfort P, Diaper MP. *Cryobiology*. 1988 Aug;25(4):285-99. doi: 10.1016/0011-2240(88)90037-5. PMID: 3136972
2. Perfusion of rabbit kidneys with cryoprotective agents. Pegg DE. *Cryobiology*. 1972 Oct;9(5):411-9. doi: 10.1016/0011-2240(72)90158-7. PMID: 4568154 No abstract available.
3. Effect of cooling and warming rate on glycerolized rabbit kidneys. Jacobsen IA, Pegg DE, Starklint H, Chemnitz J, Hunt C, Barfort P, Diaper MP. *Cryobiology*. 1984 Dec;21(6):637-53. doi: 10.1016/0011-2240(84)90223-2. PMID: 6394215
4. Flow distribution and cryoprotectant concentration in rabbit kidneys perfused with glycerol solutions. Pegg DE, Robinson SM. *Cryobiology*. 1978 Dec;15(6):609-17. doi: 10.1016/0011-2240(78)90085-8. PMID: 743886 No abstract available.
5. Perfusion of rabbit kidneys with glycerol solutions at 5 degrees C. Pegg DE, Wusteman MC. *Cryobiology*. 1977 Apr;14(2):168-78. doi: 10.1016/0011-2240(77)90137-7. PMID: 862413 No abstract available.

Response 1: We thank the reviewer for the opportunity to address this. We completely agree – we did not adequately acknowledge previous work on the topic of gradual loading in the original submission. We have added the following to the discussion, which also includes all references provided above, to address this important point (changes highlighted in yellow):

“Others have shown gradual and controlled perfusion of cryoprotective agents is required to avoid injury⁴⁸⁻⁵². We built on this previous work in the present protocol by using gradual (un)loading and the presence of osmotic counter ions.”

Question 2: The authors chose to use equivalent volumes of ethylene or propylene glycol e.g. 60 ml in their perfusate formulations. The impact of solutes on depression of freezing point or amount of ice formed at a given temperature is a molar effect of depression of freezing. The molecular masses of EG and PDiol differ by about 15% – so the concentrations of active solutes will be different in their perfusates. The authors might like to Discuss this to give readers a better understanding of the biophysics.

Response 2: The reviewer makes a great point. We have added additional information in the discussion to point out this critical difference between molecular masses of the main permeating cryoprotectants used and how this impacts depression of freezing point. The following language was added to the discussion (changes highlighted in yellow): “It should be noted that equivalent volumes of these permeating CPAs were used in the final storage solution (see Table S1), yet their molecular masses differ (EG 62.07 g/mol, PG 76.09 g/mol, GLY 92.09 g/mol). Since depression of freezing point or amount of total ice at a given temperature is a molar effect, these differences could be one factor that influenced outcomes.”

Reviewer #3: We thank the reviewer for careful consideration of the manuscript and thoughtful comments that have improved the manuscript. We provide a point-by-point response to all comments below:

Question 1: The authors should however better explain why they perform this study, as their previously published supercooling technique has already shown a clear extension of the preservation time to 4 days in rodents, and to 27h in human. What would be the benefit of exploring even deeper temperatures?

Response 1: We thank the reviewer for the opportunity to address this very important comment. Extensions of preservation duration of human livers beyond 27 hours would be required to enable global matching programs. Hence, to achieve even longer extensions of preservation, we need to descend into colder storage temperatures, where phases changes from liquid to ice can no longer be avoided. Please see the third paragraph of the introduction that has been modified to capture this important comment from the reviewer.

Question 2: I would disagree that all transplantable organs need currently a clear extension of the preservation time. What is currently more needed in many countries, is a repair strategy of injured organs, and a reliable assessment of the degree of injury before use, which however both appears limited by a static approach, with deep metabolic depression during any freezing procedure. This should be further discussed.

Response 2: The reviewer brings up another critical point that we elaborate on herein. While it is certainly true that not all transplantable organs currently ‘need’ preservation extension to reach their recipient, there are major ‘classes’ of organs that would be significantly impacted by extending preservation duration, albeit in distinct ways, as described below:

- A) *Livers that can currently reach their recipient using clinical hypothermic preservation at 4°C* - even in these cases extending preservation duration would a) reduce the cost of transplantation, b) convert from emergency to planned surgeries, and c) enable improved matching according to HLA compatibility (HLA matching is not currently possible for liver with clinical preservation duration).
- B) *Livers that are rejected during transplantation* - Up to 50% of transplanted livers could experience an acute rejection episode and new immune tolerance induction protocols are poised to eliminate rejection. However, these protocols require more time to prepare the recipient than is currently possible with clinical preservation and hence extended preservation will be required to fully realize this breakthrough achievement.
- C) *Livers that are procured but not transplanted* - 25% of livers procured for transplantation are discarded due to circumstantial factors, such donor/recipient location. These factors could be eliminated with extending preservation duration and hence would directly reduce liver discard rates.

Please see the first paragraph of the introduction that has been modified to capture this important comment from the reviewer.

Another important point surrounds the relative impact of static versus continuous perfusion modalities since perfusion provides a means to assess and revive injured organs. Importantly, it should be noted that our protocol combines a static preservation phase, in addition to machine perfusion strategies for repair and assessment (see the full protocol in Figure 1). In this way, we can achieve longer preservation durations as compared to clinical standards and supercooling, yet still leverage the unique advantages of machine perfusion such as repair and assessment. Hence, our partial freezing approach has immense synergy with machine perfusion strategies and together they have the capacity to address a multitude of

barriers in organ transplantation, as described above. Please see the second paragraph of the introduction that has been modified to capture this important comment from the reviewer.

Question 3: Why did the authors choose a model of healthy livers preserved for 24 hours? As stated, these livers are regarded to result in 100 % survival after transplantation. The applied freezing methods however did even under such conditions not reach the values of controls in every group, which is a clear hurdle, please comment on this.

Response 3: The choice of controls is of course an important consideration in the present study. The clinical limit for preservation duration of human livers is 9-12 hours, depending on the transplant center. Previous studies have repeatedly shown that the corresponding maximum preservation duration in the rodent model is 24 hours under the exact same conditions, which results in 100% transplant survival (see references 14, 32, and 33). As such, demonstrating that our partial freezing method could achieve the standards of transplanted controls would represent a high bar of achievement.

As the reviewer points out, however, the data in the original submission did not reach the values of controls in every group and we recognize this hurdle. We have done two central things to address this hurdle:

- a) Experimentally, we have improved our protocol and added new experimental groups that show significant improvements in assessment and injury parameters measured, bringing us closer to the “transplantable controls” currently used in the present study (see new Figs. 4-5, S6, S8 and corresponding results presented in “*Effects of a clinical grade oxygenator on liver viability after partial freezing*”). We improved our protocol by using a clinical grade oxygenator during machine perfusion that contains a hydrophilic polymer coating to reduce platelet adhesion and activation. While normothermic temperatures and the presence of whole blood are essential to fully realize tissue injury, recirculating whole blood may interact with the surfaces of the perfusion system, thereby triggering coagulation and inflammatory events that are not reflective of *in vivo* events.
- b) There are rich data sets available in literature on the topic of clinical *ex vivo* liver machine perfusion. As a result, in addition to our choice of control in this paper (i.e. healthy livers preserved for 24 hours), we have modified the manuscript to also contain a more in depth comparison with literature values where applicable. For example, while our maximum AST (Fig. 5b) and ALT (Fig. 5c) values for partially frozen livers at -15°C for 5 days during simulated transplantation were elevated above healthy controls, these values are nonetheless favorable as compared to literature values characteristic of ‘viable’ human livers (see new references added 46 and 47).

Question 4: Reperfusion on the isolated perfusion model is of limited value, especially if run only for 2 hours. To safely evaluate liver viability and function, much more endpoints would be needed, for example assessment of mitochondrial metabolites and energy charge, endothelial integrity, non-parenchymal cell activation, DAMP signaling, etc. In addition, I would also recommend histology staining for each type of liver cells.

Response 4: We have added the following additional endpoints to the manuscript:

- 1) Energy charge – please see Figures 2e, 3e, 4f, and S7f
- 2) DAMP signaling – please see Figure S8 that includes quantification of cfDNA, 8OHdG, TNF α , and HSP70
- 3) Histology, including H&E and IHC – please see Figs. 2g, 3g, 5e, 5f

- 4) We also included a blinded and experienced pathologist who commented specifically on important histological features such as endothelial integrity. The outcome of these analyses are summarized at the end of the following results sections: “*Effect of permeating cryoprotective agents and warm ischemic injury on liver viability after partial freezing,*” “*Effects of storage temperature and duration on liver viability after partial freezing,*” and *Effects of a clinical grade oxygenator on liver viability after partial freezing.*”

Question 5: All experiments should then be repeated in injured livers, for example in livers exposed to additional warm ischemia or steatotic livers.

Response 5: We have added a new experimental condition that evaluates the impact of warm ischemic injury on perfusion outcomes after partial freezing with propylene glycol. These data are presented in Figure S7 and the results summarized in “*Effect of permeating cryoprotective agents and warm ischemic injury on liver viability after partial freezing.*” In summary, we show that at by the end of simulated transplantation there were no statistically significant changes in all parameters measured, except for lactate.

Question 6: The introduction and the discussion should consider previous results regarding supercooling, and explain why further research with deeper temperature is needed.

Response 6: We agree with the reviewer and have included a more comprehensive and clear explanation why deeper storage temperatures in the presence of ice are needed (please see the third paragraph of the introduction). Please also see our response to Question 1 above.

Question 7: Finally, a transplant model should be used in all groups to convince on viability of frozen livers.

Response 7: Unfortunately, due to the pandemic several of our key liver transplant fellows moved on and it has not been feasible to replace them by training new fellows. Further, due to the COVID 19 pandemic, recruitment and training for such complex procedures became impossible. Instead, guided by constructive comments raised by this reviewer, we added significantly more data with exciting new findings that have improved the manuscript.

Reviewers' Comments:

Reviewer #3:

Remarks to the Author:

This is the revised version of an experimental study on partial freezing of rodent livers to extend preservation time. The authors intended to demonstrate that livers frozen for 5 days at -15°C showed favorable outcome.

Criticisms

The manuscript is well written and the authors added a significant amount of data in their corrected version. Many questions however, remain rather unaddressed:

1. It is unclear, why the demonstrated freezing method is superior in terms of outcome compared to the super cooling approach, as survival in a transplant model is not shown in this study.
2. The response that 50% of human livers are currently rejected during transplantation is wrong. Acute T cell mediated rejection rates range between 12-15 % in most programs, graft loss due to acute rejection is extreme rare. Antibody mediated rejection is even less frequent in livers, although potentially underestimated. Accordingly, HLA matching plays currently no role in liver transplantation, in contrast to infections, kidney injury, and biliary complications. Most importantly, an extension of the preservation time alone would not automatically increase liver utilization rate. For this, a reliable assessment of liver function before use is needed, which however is clearly limited by a static approach.
3. All endpoints for reperfusion injury remain focused on two hours of ex vivo reperfusion, which is far too short for concluding reliable liver viability. This is a major shortcoming, as for example the supercooling technique succeeded in three month survival of 100 % after 3 days in a transplant model, and in 58 % after 4 days.
4. The authors argue that normothermic perfusion methods become more complex for enabling ex vivo preservation for several days. However, the presented freezing technology appears not much easier, with a combination of subnormothermic perfusion, hypothermic perfusion, partial freezing, thawing, hypothermic perfusion and subnormothermic perfusion. A conclusion on the effect and on the required effort would need a head to head comparison of all available technologies.

RESPONSE TO REVIEWERS

Reviewer #3: We thank the reviewer for careful consideration of the manuscript and thoughtful comments that have improved the manuscript. We provide a point-by-point response to all comments below:

Question 1: It is unclear, why the demonstrated freezing method is superior in terms of outcome compared to the super cooling approach, as survival in a transplant model is not shown in this study.

Response 1: Supercooling of organs is a promising approach which we previously successfully demonstrated both in rat and human liver models. Nevertheless, supercooling has several limitations which we believe partial freezing can address. As we discuss in the text, supercooling is inherently limited by the depth of metabolic stasis that can be achieved. Yet, extension of preservation duration of human livers beyond 27 hours (achieved by our recent supercooling studies) would be required to enable global matching programs. Thus, alternative strategies will be required to reach lower storage temperatures and even longer preservation durations. We posit that partial freezing – as demonstrated here – can enable us to attain lower temperatures while still retaining intact tissue structure, relatively low expression of injury markers, and good recovery of important metabolic functions upon rewarming of organs before use. Additionally, partial freezing provides a more mechanically stable extracellular environment (solid) compared to supercooling especially at the lower temperatures (currently unattainable via supercooling approaches) we used in this study. We posit that the lower accessible temperatures together with the potentially higher mechanical stability could ultimately prove partial freezing as a better and more practical solution in whole organ preservation.

We had aimed to demonstrate the success of partial freezing via transplantation studies. Unfortunately, due to the pandemic several of our key liver transplant fellows moved on and it has not been feasible to replace them by training new fellows. Further, due to the COVID 19 pandemic, recruitment and training for such complex procedures became impossible. Instead, we added significantly more data with exciting new findings that have improved the manuscript. Finally, where relevant, we discuss our results as compared to literature values that utilize a transplant model (including our supercooling approach) to provide more depth and context on what the outcomes of the study could mean.

We also added the following sentence (highlighted yellow in the main text) to note the importance of an eventual clear comparison between methods - we and others develop - to find an optimally practical solution to address clinical needs:

“Each approach has advantages and disadvantages (e.g., length of preservation, thermodynamic stability, ease of operation, accessibility, etc.) that will ultimately necessitate a head-to-head comparison of all available technologies prior to clinical translation.”

Question 2: The response that 50% of human livers are currently rejected during transplantation is wrong. Acute T cell mediated rejection rates range between 12-15 % in most programs, graft loss due to acute rejection is extreme rare. Antibody mediated rejection is even less frequent in

livers, although potentially underestimated. Accordingly, HLA matching plays currently no role in liver transplantation, in contrast to infections, kidney injury, and biliary complications. Most importantly, an extension of the preservation time alone would not automatically increase liver utilization rate. For this, a reliable assessment of liver function before use is needed, which however is clearly limited by a static approach.

Response 2: We thank the reviewer for pointing this out. The percent occurrence of rejection quoted in the response was sourced from the UCSF Liver Transplant Program website. We agree the actual values are much lower, as the reviewer points out. These organ rejection rates were not quoted anywhere in the manuscript.

Further, we have softened the language which discusses the relationship between extended preservation and its direct relationship to organ utilization. Namely, we have removed the following language “[extended preservation] would directly reduce organ discard rates” from the first paragraph of the introduction and modified the first paragraph of the discussion (highlighted in yellow).

Finally, we also agree that reliable assessment is a significant issue in transplantation. We have added this caveat to the first paragraph of the introduction (below and highlighted in yellow in the main text).

“However, it should be noted that to fully transform organ allocation, utilization, and transplantation practices, an essential compliment to extensions of preservation times will be development of reliable organ assessment tools.”

We do note that while our approach does feature a static (frozen) phase, the protocol also includes a 3-hour recovery with machine perfusion (see Figure 1, step 8) that could be paired with effective assessment strategies prior to transplantation.

Question 3: All endpoints for reperfusion injury remain focused on two hours of ex vivo reperfusion, which is far too short for concluding reliable liver viability. This is a major shortcoming, as for example the supercooling technique succeeded in three month survival of 100 % after 3 days in a transplant model, and in 58 % after 4 days.

Response 3: Please see our response to Question 1 re: transplantation. In addition, to the additional endpoints added during the last revision, we now add language to the final paragraph of the discussion which emphasizes this important point by the reviewer (below and highlighted in yellow in the main text).

“Essential next steps are to confirm long-term viability with an established transplant model in rodent and swine.”

Question 4: The authors argue that normothermic perfusion methods become more complex for enabling ex vivo preservation for several days. However, the presented freezing technology appears not much easier, with a combination of subnormothermic perfusion, hypothermic

perfusion, partial freezing, thawing, hypothermic perfusion and subnormothermic perfusion. A conclusion on the effect and on the required effort would need a head-to-head comparison of all available technologies.

Response 4: We agree that a head-to-head comparison of all available technologies is the best way to make a conclusion on the required effort for each technology. We have added language to the second paragraph of introduction to emphasize this important point by the reviewer (below and highlighted in yellow in the main text).

“Each approach has advantages and disadvantages (e.g., length of preservation, thermodynamic stability, ease of operation, accessibility, etc.) that will ultimately necessitate a head-to-head comparison of all available technologies prior to clinical translation.”

Reviewers' Comments:

Reviewer #3:

Remarks to the Author:

This is the revised version of a manuscript on partial freezing of rodent livers. The authors tested in this experimental study the effect of 5 days liver preservation at -15°C , compared to conventional cold storage. The authors point out that this approach potentially extends preservation substantially, allowing theoretically future organ banking.

While the manuscript has certainly been improved and reads well, the major shortcoming of this study remains the assessment of liver viability after preservation. The authors explain in their point-to-point reply that a transplant model could not be implemented due to the Covid epidemic and the difficulty to train new fellows, which I understand.

The model, which is consecutively used for simulating transplantation, is the isolated perfused rat liver with however a very short reperfusion time, e.g. two hours. I would have expected, that authors add some experiments, for example in the 5 day preservation group, with an ex vivo reperfusion of at least 6 hours, demonstrating convincing endpoints which show no or only small differences compared to controls.

Instead, the authors report, that adenylate energy charge and bile flow were extremely low, e.g. < 0.1 , < 5 ml, respectively, in the 5 day preservation group (PG -15°C), besides an increasing lactate level up to 6mM (Figure 3). I realize that these parameters were not significantly different from the 1 day -15 preservation group, but it remains unclear in terms of this data, whether these livers are viable. The addition of biomarkers, e.g. albumin, LDH, FMN, and IL8 (Figure S9E,F) is from my point of view not very helpful, as also in these markers, controls perform different from the 5 day preservation group, and the sample size appears very small per marker ($n=3-4$).

Based on this, I would not agree with the statement that livers, partially frozen at -15°C for 5 days, showed favorable outcomes, as compared to viable livers.

RESPONSE TO REVIEWERS

Reviewer #3: We thank the reviewer for careful consideration of the manuscript. We provide a point-by-point response to all comments below:

Question 1: While the manuscript has certainly been improved and reads well, the major shortcoming of this study remains the assessment of liver viability after preservation. The authors explain in their point-to-point reply that a transplant model could not be implemented due to the Covid epidemic and the difficulty to train new fellows, which I understand.

The model, which is consecutively used for simulating transplantation, is the isolated perfused rat liver with however a very short reperfusion time, e.g. two hours. I would have expected, that authors add some experiments, for example in the 5 day preservation group, with an *ex vivo* reperfusion of at least 6 hours, demonstrating convincing endpoints which show no or only small differences compared to controls.

The addition of biomarkers, e.g. albumin, LDH, FMN, and IL8 (Figure S9E,F) is from my point of view not very helpful, as also in these markers, controls perform different from the 5 day preservation group, and the sample size appears very small per marker (n=3-4). Based on this, I would not agree with the statement that livers, partially frozen at -15°C for 5 days, showed favorable outcomes, as compared to viable livers.

Response 1: It looks like we may have simply misunderstood the original question from the reviewer since it is absolutely feasible for us to add a longer normothermic reperfusion time at 6 hours.

In the original comments from the reviewer, it was noted that the normothermic perfusion time was a shortcoming, although we felt this was discussed in the context of transplantation studies (i.e., months of survival). Indeed, the specific expectation of the reviewer to add 6 hours of perfusion was not explicitly stated in their original comments. Moreover, it is standard for simulated transplantation assessment, consisting of normothermic perfusion in the presence of whole blood, to be performed on the order of ~2 hours since fulminant injury occurs rapidly. Finally, we should note that our complete protocol consists of viability endpoints collected post-preservation during 3 hours of subnormothermic perfusion (supplementary material Fig. S4, S5, S6) and 2 hours of simulated transplantation assessment (normothermic perfusion in the presence of blood or reperfusion). As such, we had presented viability indices for a total perfusion time of 5 hours post freezing/preservation. For all of these reasons, we misinterpreted the timescale of the question and instead responded with the addition of well-known endpoints to address the core of the reviewer's question, which is viability of the grafts.

In this rejection appeal, we add experiments that use a 6-hour simulated transplantation in the presence of whole blood to be re-considered by the third reviewer. As requested by the reviewer, we performed these experiments with the 5-day preservation group and controls. This new data is presented in Figure S10. In addition, we also present literature values that are used in clinical trials for *ex vivo* liver perfusion to make go/no-go decisions to transplant. While perfusion-based criteria to define liver viability can vary across countries/institutions, the VITTAL (viability testing and transplantation of marginal livers) clinical trial defines livers as suitable for transplantation when the following criteria are met: organs metabolizing lactate to ≤ 2.5 mmol/L within 4 hours of the perfusion commencing in combination with two or more of the following parameters - bile production, metabolism of glucose, a hepatic arterial flow rate ≥ 150 mL/min and a portal venous flow rate ≥ 500 mL/min, a pH ≥ 7.30 and/or maintain a homogeneous perfusion [please see reference 54 for more info]. To succinctly compare important experimental endpoints to our controls as well define what these values mean in the context of clinical trial data, we present Table 1 (also included below for convenience).

To summarize Table 1- all livers that were stored for 5 days at -15°C showed endpoint values after 6 hours simulated transplantation that were either not significantly different from controls or are within the range that is considered transplantable based on clinical criteria. The only exception was pH that decreased as a function of perfusion for the 5-day frozen livers, although importantly not all criteria need to be met for livers to be considered transplantable (only 2 of 5 criteria need to be met and our livers meet 4 of 5 criteria). Taken together, we argue that livers partially frozen at -15°C for 5 days show favorable outcomes, as compared to viable livers.

Table 1: Rodent livers that were partially frozen at -15°C for 5 days are viable using our 6-hour perfusion model as compared to transplantable controls and/or clinical liver perfusion data, representing a 5-fold extension of preservation duration. Controls are livers that were stored at 4°C in the University of Wisconsin solution for 24 hours. Perfusion-based metrics that are used in clinical *ex vivo* liver perfusion (EVLV) to define liver suitability for transplantation include lactate falling below 2.5 mM in combination with 2 out of 5 other indicators (bile production, glucose metabolism, venous/arterial flow rates, pH, and uniformity of perfusion)⁵⁴. Rodent liver data and statistical results presented are from the final, 6-hour time point of our simulated transplantation model, unless otherwise stated.

Viability Index	Partial Freezing @ -15°C for 5 days	Hypothermic Control @ 4°C for 1 day	Clinical EVLP
Lactate	All livers metabolized lactate. Mean 2.65 ± 0.77 mM at 4h and 1.843 ± 0.97 mM at 6h.	No difference with frozen livers.	Organs metabolizing lactate to ≤ 2.5 mmol/L at 4h [54].
Bile Production	All livers produced bile.	Total accumulation was higher than frozen livers.	Livers are producing bile [54].
Glucose	All livers metabolized glucose.	No difference with frozen livers.	Metabolism of glucose [54].
pH	pH decreased throughout perfusion.	pH was higher than frozen livers.	A pH ≥ 7.30 [54].
Flow/Resistance	All livers reached target flows.	No difference with frozen livers.	A hepatic arterial flow rate ≥ 150 mL/min and a portal venous flow rate ≥ 500 mL/min [54].
Uniformity	Homogeneous perfusion achieved.	No difference with frozen livers.	Maintain a homogeneous perfusion [54].
Injury Markers			
ALT	Mean 768 ± 206 U/L.	ALT lower than frozen livers.	ALT $\leq \sim 3000$ U/L [46]. Range 152-1460 U/L [55].
AST	Mean 1270 ± 430 U/L.	AST lower than frozen livers.	Range 227-9200 U/L [55].

Reviewers' Comments:

Reviewer #3:

Remarks to the Author:

This is the revised version of a paper on partial freezing of rat livers. The authors added another experimental group with 6 hour ex vivo reperfusion to prove that the 5d preservation group resulted in viable livers, compared to controls.

While I acknowledge highly the three additional performed Experiments in each group, the authors base their evaluation on viability criteria of the VITTAL trial. These criteria include lactate levels, and bile production, which are however no reliable parameters of liver function (Watson et al, AJT 2018, Eshmuninov et al, Nature Biotechnology 2020). For example, livers, which produce bile are not necessarily functioning after implantation, and livers which clear lactate can be highly necrotic. Recent data show in this respect that bile quality is more informative than bile quantity. In addition, the amount of bile on Figure S10b is extremely low compared to controls, and liver enzymes were 1270 (AST) and 767 (ALT). This should be clearly stated as a limitation. In addition, the conclusion that these livers are not significantly from controls or within the range of controls should be rephrased.

RESPONSE TO REVIEWERS

Reviewer #3: We thank the reviewer for careful consideration of the manuscript. We provide a point-by-point response to all comments below:

Question 1: While I acknowledge highly the three additional performed Experiments in each group, the authors base their evaluation on viability criteria of the VITTAL trial. These criteria include lactate levels, and bile production, which are however not reliable parameters of liver function (Watson et al, AJT 2018, Eshmuninov et al, Nature Biotechnology 2020). For example, livers, which produce bile are not necessarily functioning after implantation, and livers which clear lactate can be highly necrotic. Recent data show in this respect that bile quality is more informative than bile quantity. In addition, the amount of bile on Figure S10b is extremely low compared to controls, and liver enzymes were 1270 (AST) and 767 (ALT). This should be clearly stated as a limitation.

Response 1: Watson et al (AJT 2018) and Eshmuninov et al (Nature Biotechnology 2020) are referenced as #65 and 12, respectively, in the main text. We have stated cumulative bile and liver enzyme levels as a limitation throughout the manuscript in the following locations:

- 1) In the last sentence of the abstract, line 42-44.
- 2) When outlining the results, we note that even though all livers produced bile, the cumulative total was lower than controls (line 305) and that liver enzymes were elevated above controls (line 310-314).
- 3) In the discussion, we comprehensively discuss bile and transaminase levels as a limitation on line 360-366. This section is copy-pasted below for convenience.

“In contrast to lactate and oxygen consumption, all frozen livers produced significantly less bile and released significant more transaminases after freezing, as compared to controls. Preclinical studies showed that reduced bile production and elevated transaminase levels negatively correlate to transplant survival^{13,61}. In the clinical setting, bile production and low transaminase levels during NMP are considered favorable however not necessary for safe transplantation of the graft if other parameters are viable^{41,54,65}. Nonetheless, in the present study we consider these as important signs of hepatocellular injury after freezing that need to be addressed.”

- 4) Finally, on line 441-443 we re-iterate the limitation of bile and liver enzymes.

Question 2: In addition, the conclusion that these livers are not significantly from controls or within the range of controls should be rephrased.

Response 2: Please note that we did not intend to imply that all our frozen livers were within the range of controls. Instead, we intended to indicate that results fell into two categories: i) those that were not statistically different from controls versus ii) values that were within an acceptable range based on human liver studies, despite being statistically different from controls. We provide references 46, 54, and 55 that indicate acceptable ranges for human livers.

We have now re-phrased it as follows (line 438-440): “after a 6-hour simulated transplantation, all livers that were stored for 5 days at -15°C showed endpoint values that can be described as either of the following: i) were not significantly different from controls or ii) considered acceptable based on clinical criteria.”